# Validation of MUSES NH₃ observations from AIRS and CrIS against aircraft measurements from DISCOVER-AQ and a surface network in the Magic Valley

**Karen E. Cady-Pereira**[1], **Xuehui Guo**[2], **Rui Wang**[3], **April B. Leytem**[4], **Chase Calkins**[1], **Elizabeth Berry**[1], **Kang Sun**[5,6], **Markus Müller**[7], **Armin Wisthaler**[7], **Vivienne H. Payne**[8], **Mark W. Shephard**[9], **Mark A. Zondlo**[2], and **Valentin Kantchev**[8,10]

[1]Atmospheric and Environmental Research Inc., Lexington, MA, USA

[2]Department of Environmental Sciences, University of Virginia, Charlottesville, VA, USA

[3]Department of Civil and Environmental Engineering, Princeton University, Princeton, NJ, USA

[4]United States Department of Agriculture-Agricultural Research Service, Kimberly, ID, USA

[5]Department of Civil, Structural and Environmental Engineering, University at Buffalo, Buffalo, NY, USA

[6]Research and Education in eNergy, Environment and Water (RENEW) Institute, University at Buffalo, Buffalo, NY, USA

[7]Institute for Ion Physics and Applied Physics, University of Innsbruck, Innsbruck, Austria

[8]Jet Propulsion Laboratory, California Institute of Technology, Pasadena, CA, USA

[9]Environment and Climate Change Canada, Toronto, ON, Canada

[10]Instrument Software and Science Data Systems, Pasadena, CA, USA

**Correspondence:** Karen E. Cady-Pereira (kcadyper@aer.com)

**Abstract.** Ammonia is a significant precursor of $PM_{2.5}$ particles and thus contributes to poor air quality in many regions. Furthermore, ammonia concentrations are rising due to the increase of large-scale, intensive agricultural activities, which are often accompanied by greater use of fertilizers and concentrated animal feedlots. Ammonia is highly reactive and thus highly variable and difficult to measure. Satellite-based instruments, such as the Atmospheric Infrared Sounder (AIRS) and the Cross-Track Infrared Sounder (CrIS), have been shown to provide much greater temporal and spatial coverage of ammonia distribution and variability than is possible with in situ networks or aircraft campaigns, but the validation of these data is limited.

Here we evaluate MUSES (multi-spectra, multi-species, multi-sensors) ammonia retrievals from AIRS and CrIS against ammonia measurements from aircraft in the California Central Valley and in the Colorado Front Range. These are small datasets taken over high-source regions under very different conditions: winter in California and summer in Colorado. Direct comparisons of the surface values of the retrieved profiles are biased very low in California ($\sim 40$ ppbv) and slightly high in Colorado ($\sim 10$ ppbv) TS1. This bias appears to be primarily due to smoothing error, since applying the instrument operator effectively reduces the bias to zero; even after the smoothing error is accounted for, the average uncertainty at the surface is in the 20 %–30 % range. We also compare 3 years of CrIS ammonia against an in situ network in the Magic Valley in Idaho We show that CrIS ammonia captures both the seasonal signal and the spatial variability in the Magic Valley, although it is biased low here also. In summary, this analysis substantially adds to the validation record but also points to the need for more validation under many different conditions and at higher altitudes.

## 1   Introduction

Ammonia ($NH_3$) is one of the most common forms of reactive nitrogen and the primary alkaline gas in the atmosphere. Intended and unintended releases of $NH_3$ into the environ-

ment over the past century have significantly altered the natural nitrogen cycle (Erisman et al., 2008), so that the current emission levels of NH$_3$ are about 4 times higher than in previous centuries (Battye et al., 2017). The main sources of NH$_3$ are agricultural emissions, namely, from livestock raising and fertilizer application (EDGAR [database] – Emissions Database for Global Atmospheric Research, 2016), which account for 80 % of all emissions globally (Sutton et al., 2013; Behera et al., 2013). There are also some locally or seasonally significant sources of NH$_3$, the most notable being biomass burning events, which can generate large amounts of NH$_3$ (Coheur et al., 2009; Whitburn et al., 2015, 2016). In urban areas automobiles with three-way catalytic converters (Sun et al., 2017) can be a major source of NH$_3$. Nowak et al. (2012) estimate that in the Los Angeles Basin cars contribute as much as 50 % of the total NH$_3$ emissions.

NH$_3$ is the dominant base in the atmosphere, and it plays a significant role in the formation of fine particulate matter (PM$_{2.5}$) (e.g., Aneja et al., 2003), which can penetrate deep into the lungs and severely impact the respiratory and circulatory systems (Pope et al., 2009). Paulot and Jacob (2014) have shown that the costs linked to the health impacts of NH$_3$ associated with food production for export in the United States offset half the revenue from these exports. Long-term exposure to ambient PM$_{2.5}$ is the leading environmental risk factor for premature mortality worldwide, leading to an estimated 2.5–3.4 million premature deaths annually (Cohen et al., 2017), 20 % of which is estimated to stem from NH$_3$ emissions (Lelieveld et al., 2015). NH$_3$ emissions are regulated by the European Union (EU) and it is a criteria pollutant in Canada, but not yet in the US. However, the Environmental Protection Agency (EPA) has published established regulations (https://www.nsrlaw.com/single-post/2017/06/19/EPA-NSR-Chief-Outlines-NSR-Changes-at-2017-AWMA-Conference, last access: 24 August 2016) mandating that every state must set area-specific significant emission rates (SERs) for NH$_3$. Since the emissions of NH$_3$ are a key factor in the formation of PM$_{2.5}$, reducing emissions can be an effective path to reduce air pollution (Liu et al., 2021).

Given the rapid growth of industrial-scale agriculture (e.g., increase in egg, milk and meat consumption), especially in Asia (e.g., Xu et al., 2016), NH$_3$ emissions are projected to increase greatly over the next few decades in many parts of the world. The reduction of NO$_x$ emissions due to more stringent controls will reduce the contribution of NO$_x$ to the deposition of reactive nitrogen, but Paulot et al. (2013) suggest that an increase in NH$_3$ emissions will likely compensate for this reduction. NH$_3$ and its derivatives are also quickly deposited in the ecosystems, increasing their eutrophication and reducing biodiversity (Erisman et al., 2008).

There is thus growing recognition that NH$_3$ is an important pollutant, and that it will likely play a greater role in air quality and ecosystem health over the next few decades, due to both the essential role NH$_3$ plays in feeding the world's population, and to the fact that the atmospheric emission potential of NH$_3$ is directly linked to increasing temperatures (Skjøth and Geels, 2013; Sutton et al., 2013). However, in situ measurements remain a challenge. NH$_3$ is easy to detect, but it is hard to measure accurately, especially for concentrations below 10 ppbv (von Bobrutzki et al., 2010). There are many in situ techniques used to detect atmospheric NH$_3$ with varying time resolution and precision, but the main issue affecting precision is the inlet rather than the instrument. NH$_3$ is sticky, and thus it is challenging to get it into a given instrument quantitatively and quickly (Roscioli et al., 2016; Pollack et al., 2019). This feature is critical for characterizing the abundance of NH$_3$ in the background atmosphere, for making measurements of NH$_3$ fluxes, and for deploying instruments on aircraft. New open-path sensors avoid this issue, but they cannot be deployed in many situations (e.g., Berkhout et al., 2017; Müller et al., 2014). Consequently, the emissions of NH$_3$ outside of a limited set of well-instrumented locations remain poorly constrained, reducing the accuracy with which models can represent concentrations and variability. The high spatial and temporal variability of NH$_3$ (surface values can range from less than 0.1 to 200 ppbv or more) exacerbates the lack of continuous, spatially well-sampled data over extensive regions. This also contributes to bottom–up inventories often underestimating emissions due to scaling difficulties (Nowak et al., 2012).

Satellite data, despite having their own uncertainties, provide by virtue of their spatial and temporal density, another option for quantifying these emissions. Currently there are multiple NH$_3$ datasets, with varying data record lengths and spatial coverage, obtained from the following instruments: the three Infrared Atmospheric Sounding Interferometer (IASI) instruments flying in a 09:30 orbit, the Greenhouse Gases Observing Satellite (GOSAT) in a 13:30 orbit, along with the Tropospheric Emission Spectrometer (TES), the Atmospheric Infrared Sounder (AIRS), and the three Cross-Track Infrared Sounder (CrIS) instruments, all flying in a 13:30 orbit. The data obtained from these instruments have had numerous applications. Multiple studies (Van Damme et al., 2015a; Shephard et al., 2011; Shephard and Cady-Pereira, 2015; Warner et al., 2016; Shephard et al., 2020; Wang et al., 2021) have shown that NH$_3$ measurements from infrared sensors capture NH$_3$ hotspots, such as the Indo-Gangetic Plain, eastern China and the American Midwest, as well as the expected regional seasonal variability and fire activity. Warner et al. (2017) used retrievals from AIRS to show definite positive trends in NH$_3$ concentrations over the US, the EU and China, which the authors ascribe to declines in SO$_2$ and NO$_2$ emissions in all three regions due to more stringent controls. Van Damme et al. (2018) used nearly a decade of IASI (ANNI-NH3-v2.1R-I) data to show that the emissions listed in the EDGAR (EDGAR, 2016) inventory for large-source regions were wrong by as much as a factor of 3; furthermore, emissions from smaller sources were often underestimated by an order of magni-

tude. Dammers et al. (2019) found similar results using CrIS (CFPR-v1.0) and IASI (ANNI-v2.2) $NH_3$ data. Zhu et al. (2013) have demonstrated that Tropospheric Emission Spectrometer (TES) $NH_3$ data over North America in the 2006–2009 period, although relatively sparse, could be used in an inverse modeling framework to constrain emissions sufficiently in order to improve agreement between GEOS-Chem output and surface measurements from the National Atmospheric Deposition Program (NADP) Ammonia Monitoring Network (AMoN) network (https://nadp.slh.wisc.edu/networks/ammonia-monitoring-network/, last access: TS2). Using $NH_3$ measurements from CrIS (multi-spectra, multi-species, multi-sensors: MUSES) and $NO_2$ measurements from TROPOMI, Cao et al. (2022) demonstrated that $NH_3$ emissions decreased substantially over downtown Los Angeles during the 2019 March COVID-19 lockdown; this result is in agreement with the conclusion from Nowak et al. (2012) that in urban areas traffic can be a major source of $NH_3$ and consequently greatly increase exposure to $PM_{2.5}$.

Yet, in spite of the increasing use of $NH_3$ data from space-based instruments, validation of these data remains rather limited. Sun et al. (2015) compared a small set of $NH_3$ total columns from the TES instrument against columns derived from surface and aircraft measurements during the NASA Deriving Information on Surface conditions from Column and Vertically Resolved Observations Relevant to Air Quality (DISCOVER-AQ) California 2013 campaign, and found small differences (less than 6 %) and high correlation ($R = 0.82$); however, note that TES, which is no longer operational, had much higher spectral resolution ($0.06\,cm^{-1}$) and thus greater sensitivity to surface $NH_3$ and less interference from water vapor than the infrared sensors (AIRS, CrIS, IASI) currently providing data for $NH_3$ retrievals. Shephard et al. (2015) compared TES profiles against aircraft measurements taken during the 2013 Joint Canada–Alberta Implementation Plan for Oil Sands Monitoring (JOSM) campaign and showed that the TES profiles were unbiased. Warner et al. (2016) compared four $NH_3$ retrievals from AIRS against aircraft profiles obtained during DISCOVER-AQ California and found good qualitative agreement. Dammers et al. (2017) compared 218 IASI (IASI-LUT and IASI-NN) and CrIS (CFPR-v1.0) total columns and CrIS profiles against corresponding data from ground-based Fourier transform infrared (FTIR) observations at seven FTIR sites in the Network for the Detection of Atmospheric Composition Change (NDACC): the FTIR and CrIS total columns from the combined data were well correlated ($R = 0.77$) and mainly unbiased. Correlations at the individual sites ranged from 0.28 (Mexico City) to 0.86 (Bremen).

Van Damme et al. (2015b) carried out what is likely the most extensive evaluation of $NH_3$ measured from space, comparing IASI (IASI-LUT) $NH_3$ against data from six different monitoring networks in North America, Europe, Africa and China and from the California Research at the Nexus of Air Quality and Climate Change (CalNex) campaign in California. Most of the data from the surface networks were provided on bi-weekly or monthly scales: when IASI columns were converted to surface concentrations and averaged over the corresponding time period, they showed qualitative agreement in space and time with the surface data. The correlations in general were not high, although still significant ($R = 0.25$–0.49). Recently Guo et al. (2021) (hereafter Guo2021) compared $NH_3$ columns from IASI (ANNI-v3) with integrated profiles obtained from aircraft data during the Colorado 2014 DISCOVER-AQ campaign: the IASI columns were unbiased and significantly correlated ($R = 0.57$). Guo2021 do point out that the instruments currently used to measure $NH_3$ from aircraft have large uncertainties due to limited accuracy and slow response to changing $NH_3$ concentrations.

To a varying degree all the aforementioned studies cite the same factors that complicate the validation of satellite $NH_3$ products:

- Different measurement time scales (weeks or days vs. instantaneous), especially for surface networks.

- High in situ instrument noise.

- Validation results are strongly influenced by local atmospheric conditions and the vertical distribution of $NH_3$, which highlights the need for further validation campaigns under diverse conditions.

- Sub-pixel inhomogeneity due to the high spatial–temporal variability of $NH_3$ driven by its short lifetime; thus the point data from an in situ instrument will only be partially correlated with the pixel scale data obtained from a satellite instrument.

Our objective is to add to the validation record at the single-pixel scale with retrievals from L1B radiances from both the AIRS and CrIS instruments. The retrieved profiles here are obtained with the MUSES (Fu et al., 2013, 2016, 2018) algorithm, which provides profiles, total columns and uncertainty estimates, all of which can also be evaluated against in situ data. AIRS and CrIS $NH_3$ will be compared against PTR-MS (proton-transfer-reaction mass spectrometer) data from the P-3B aircraft flown during DISCOVER-AQ campaigns in California and Colorado. Warner et al. (2016) also compared AIRS with DISCOVER-AQ, but their retrievals used cloud-cleared radiances and extended over nine AIRS pixels ($\sim 45\,km$ footprints at nadir). Using single-pixel radiances provides several advantages over "cloud-cleared radiances": the propagation of uncertainties from the radiances is simpler (see Sect. 2.2) and the retrieved information is obtained on smaller spatial scales, which is important for $NH_3$ (see Sect. 5). This will be the first comparison of single-pixel $NH_3$ profiles from either AIRS or CrIS against aircraft data. Aircraft campaigns are valuable in that they profile the vertical distribution of $NH_3$, allowing us to evaluate the performance of retrieval algorithms and to provide models with

more realistic profiles; however, they are by nature limited in their temporal coverage. In order to test the capability of MUSES $NH_3$ to capture temporal and spatial variability over an extended period, surface-level CrIS $NH_3$ concentrations will also be evaluated against 3 years of data from a small monitoring network in the Magic Valley in Idaho. Section 2 briefly reviews the $NH_3$ retrieval algorithm, Sect. 3 gives an overview of the instruments, Sect. 4 presents the analysis of the DISCOVER-AQ data and Sect. 5 follows with the analysis of the Magic Valley data; finally, Sect. 6 summarizes our conclusions and discusses future work.

## 2  MUSES $NH_3$ retrieval algorithm and data

### 2.1  MUSES algorithm

The first nadir retrievals of $NH_3$, obtained using a prototype retrieval with data from the TES instrument (Beer et al., 2008), exploited the $NH_3$ $\nu_2$ vibrational band between 960 and 970 cm$^{-1}$ to demonstrate that TES could measure the variability in $NH_3$ along a transect in China. Shephard et al. (2011) implemented a full optimal estimation (OE) approach (Rodgers, 2000), which sought to reduce the difference between the measured CrIS radiances in the $\nu_2$ band and the calculated radiances from a radiative transfer model (Moncet et al., 2008). Before the OE algorithm was run, an a priori profile was chosen from one of three possible profiles (Fig. S1 in the Supplement), representing background, moderate and enhanced $NH_3$ concentrations. These profiles were derived by binning global distributions of $NH_3$ (Shephard et al., 2011) from the chemical transport model GEOS-Chem (Zhu et al., 2013). The profile is selected by applying an online/offline brightness temperature (BT) difference test centered around the 967 cm$^{-1}$ line. The OE algorithm is then run as a refinement step, in which the a priori and the initial guess profiles are identical except for the background profile, for which the moderate profile is chosen as the initial guess, in order to provide Jacobians with some sensitivity.

The algorithm developed for the TES $NH_3$ retrievals has since been adapted with minor changes for CrIS (Shephard and Cady-Pereira, 2015; Shephard et al., 2020) and AIRS (this paper). The spectral retrieval window and the frequencies for the online/offline BT test were slightly modified for the CrIS and AIRS spectral resolutions, and a preliminary retrieval step to adjust the surface temperature and emissivity was introduced. This algorithm forms the core of the $NH_3$ component of the MUSES software used here and also of the CrIS Fast Physical Retrieval (CFPR) code, whose product has been used in a number of previous studies (e.g., Shephard and Cady-Pereira, 2015; Dammers et al., 2017; Shephard et al., 2020; Cao et al., 2022; Marais et al., 2021). The two products have much in common (the same spectral microwindows, a priori selection, constraint matrices and forward model), but obtain temperature and water profiles and

surface properties from different sources and use different software to carry out the optimal estimation (Table S2 in the Supplement). Preliminary comparisons have shown good agreement on average between the two algorithms (Fig. S2), but a full comparison is beyond the scope of this paper, as the objective here is the validation of the MUSES AIRS and CrIS $NH_3$ retrievals.

The MUSES algorithm is an end-to-end optimal estimation process that provides a complete characterization of the parameters involved in the radiative transfer processes in the infrared region, using a multi-step approach. Before the $NH_3$ retrieval step is reached, the atmosphere has been well characterized by the previous retrieval steps: temperature and water vapor profiles, surface properties and cloud absorption are thus known and can be accounted for in the $NH_3$ retrieval, significantly reducing errors from radiatively interfering species. Other species, such as carbon monoxide and ozone, are also retrieved in separate steps. This ensures that the atmospheric state is derived using the same forward model and radiance data that are used in the NH3 retrieval, reducing possible sources of error. Since cloud optical depth is retrieved, cloud-clearing algorithms (Susskind et al., 2003) are not needed and retrievals can be performed on every pixel, or field of view (FOV), rather than on the 9-pixel field of regard (FOR). This allows for retrievals from AIRS with a 15 km rather than 45 km minimum footprint, which was the resolution for the earlier $NH_3$ retrievals from AIRS obtained by Warner et al. (2016) using cloud-cleared radiances.

MUSES uses the optimal spectral sampling (OSS) model (Moncet et al., 2008) as its forward model; OSS is a fast and accurate radiative transfer method designed specifically for the modeling of radiances measured by sounding radiometers in the infrared, although it is applicable throughout the microwave, visible, and ultraviolet regions.

Since the retrieval is non-linear, an a priori constraint is used for estimating the true state (Bowman et al., 2006). If the estimated (retrieved) state is close to the actual state, then the estimated state can be expressed in terms of the actual state through the linear retrieval (Rodgers, 2000):

$$\hat{x} = x_a + A(x - x_a) + Gn + GK_b(b - b_a), \tag{1}$$

where $\hat{x}$, $x_a$, and $x$ are the retrieved, a priori, and the "true" state vectors, respectively, $G$ is the gain matrix, $b$ is the vector that contains parameters not retrieved in the current step and $b_a$ the a priori values for these parameters if they are retrieved in another step.

The averaging kernel, $A$, describes the sensitivity of the retrieval to the true state:

$$A = \frac{\partial \hat{x}}{\partial x} = \left(K^T S_n^{-1} K + S_a^{-1}\right)^{-1} K^T S_n^{-1} K = GK, \tag{2}$$

where $S_n$ is the instrument noise covariance matrix, and $S_a$ is the a priori covariance matrix for the retrieval. The Jacobian, $K$, is the sensitivity of the forward model radiances to

the true state vector, $K = \partial L / \partial x$. The rows of $\mathbf{A}$ are functions with a finite width corresponding to the vertical resolution of the retrieved parameter. The sum of each row of $\mathbf{A}$ provides a measure of retrieval information that comes from the measurement (Rodgers, 2000) at the corresponding altitude, provided that the retrieval is quasi-linear. The trace of the averaging kernel matrix gives the number of degrees of freedom for signal (DOFS) from the retrieval.

The total a posteriori error covariance matrix $\mathbf{S}_x$ for a given retrieved parameter $\hat{x}$ is given by

$$\mathbf{S}_x = (\mathbf{A} - \mathbf{I})\,\mathbf{S}_a(\mathbf{A} - \mathbf{I})^{\mathrm{T}} + \mathbf{G}\mathbf{S}_n\mathbf{G}^{\mathrm{T}} + \mathbf{G}\mathbf{K}_b\mathbf{S}_b(\mathbf{G}\mathbf{K}_b)^{\mathrm{T}}, \quad (3)$$

where $\mathbf{S}_b$ is the expected covariance of the non-retrieved parameters. The total error (or uncertainty) for a retrieved profile is expressed as the sum of: (i) the smoothing errors (first term on the right-hand side), i.e., the uncertainty due to unresolved fine vertical structure in the profile; (ii) the measurement errors (second term) originating from random noise in the spectrum; and (iii) the systematic errors (last term) due to uncertainties in the forward model parameters not retrieved in the $NH_3$ step, some of which are constant and some of which change from retrieval to retrieval (Worden et al., 2006). The MUSES $NH_3$ retrieval step includes the estimated errors in water vapor and temperature in the systematic errors. For the retrieved profiles used in this study the measurement error ranged from 3 % to 23 %, the systematic errors mainly from 1 % to 60 %, and the smoothing errors from 24 % to 130 %. Example retrieved profiles and corresponding errors are shown in Fig. S3. By providing the expected error covariance and the averaging kernels, this approach facilitates the use of the retrieved profiles in inverse modeling efforts, since both terms are used to weight the information coming from the retrieval. The error covariance gives users an uncertainty estimate for each retrieved profile, which can be utilized to screen the data or be included in a statistical analysis. Furthermore, the estimated uncertainty derived from the error covariance can be compared with measured uncertainties, obtained by calculating the spread of the differences between satellite and in situ data, as will be shown in Sects. 4.1 and 4.2; this analysis can indicate whether there are error terms missing from the optimal estimation formulation. Note that the estimated error cannot account for sampling errors, i.e., differences between the air masses sampled by the satellite and by the in situ instruments, or for sub-pixel variability.

Rodgers and Connor (2003) presented a method for comparing satellite profiles of trace gases with limited vertical resolution with in situ profiles obtained on a much finer grid. This approach is often described as "applying the instrument operator" or "applying the averaging kernel". It attempts to estimate how the space-based instrument would "see" an in situ profile by applying the equation below to the in situ data:

$$X = X_{\mathrm{apriori}} + A\left(X_{\mathrm{aircraft}} - X_{\mathrm{apriori}}\right). \quad (4)$$

The estimated profile $X$ has been smoothed by the operator, simulating the smoothing due to the coarser resolution of the satellite observation. When $X$ is compared with the satellite observations it is assumed that the smoothing error has been accounted for and can be ignored, which is not the case when satellite observations are compared directly with measured profiles; the remaining errors will be due to instrument noise and temporal and spatial sampling differences; the latter can be especially large for $NH_3$, due to its large variability, as was discussed earlier. We will follow the Rodgers and Connor (2003) approach as described earlier, as it is the optimal method for taking into account the sensitivity of the instruments; however, as a way of introducing the data while also demonstrating the pitfalls of not considering the different vertical resolution of the satellite and aircraft data, we will also show simple differences between the aircraft and satellite data.

A note on applying the operator: AIRS and CrIS profiles extend to the top of the atmosphere, while the aircraft profiles used here rarely go above 700 hPa in California and 500 hPa in Colorado. We have extended the aircraft profiles by blending in the MUSES a priori profile above the top of the aircraft profile, then applied the instrument operator to these extended profiles.

## 2.2 MUSES data

### 2.2.1 AIRS single-pixel $NH_3$ retrievals

AIRS is a nadir-viewing, scanning thermal infrared (TIR) spectrometer launched on board the Aqua satellite on 4 May 2002, into a sun-synchronous polar orbit at an altitude of 705 km with a 01:30 local solar time (LST) Equator in the descending node and 13:30 LST in the ascending node (Aumann et al., 2003). The daytime overpass is an ideal time for $NH_3$ retrievals, as thermal contrast is high and emissions are usually peaking, driven by higher temperatures. AIRS measures the thermal radiance between 3 and 12 µm with a spectral resolution of $\sim 0.75\,\mathrm{cm}^{-1}$ and a noise level of $\sim 0.15\,\mathrm{K}$ at 270 K (Zavyalov et al., 2013) in the $970\,\mathrm{cm}^{-1}$ $NH_3$ absorption window. A single AIRS FOV has a circular footprint with $\sim 15\,\mathrm{km}$ diameter at nadir and the AIRS swath width is $\sim 1650\,\mathrm{km}$, which enables near global coverage twice daily.

### 2.2.2 CrIS single-pixel $NH_3$ retrievals

CrIS is a Fourier transform infrared radiometer (FTIR) launched on the Suomi National Polar Orbiter Preparatory (SNPP) platform in October 2011 into sun-synchronous orbits (824 km altitude) with the same LST crossing times as AIRS (13:30 and 01:30 LST). CrIS was also deployed on the Joint Polar Satellite System (JPSS-1) in November 2017 and on JPSS-2 in November 2022, but in this paper only CrIS data from the SNPP platform have been used. CrIS is a cross-track scanning instrument with a 2200 km swath width

($\pm 50°$) with 14 km circular pixels (at nadir), a spectral resolution of $0.625\,\mathrm{cm}^{-1}$ and low spectral noise ($\sim 0.04\,\mathrm{K}$ at 270 K) (Zavyalov et al., 2013) in the $NH_3$ spectral window. CrIS also provides twice daily global coverage. Note that in this paper there is a data gap in the SNPP-CrIS record between March and June 2019, corresponding to a malfunction in the electronics during that time.

While both AIRS and CrIS have the same equatorial crossing time, this does not imply that both instruments are observing the same location at the same time at the same angle. For example, on 22 January 2013, CrIS flew almost directly over the Central Valley around 21:00 UTC, while AIRS had its closest observation 15° to the west around 22:00 UTC, near the edge of its swath. Therefore, the set of aircraft profiles co-located with each instrument is not identical.

## 3 Data

### 3.1 Aircraft data

The NASA DISCOVER-AQ (Crawford and Pickering, 2014) campaigns were designed to validate collocated satellite observations of atmospheric pollutants over four regions in the United States: (Maryland/Washington DC, California, Texas and Colorado), but $NH_3$ was only measured in California and Colorado. Vertical profiles of $NH_3$ were obtained by the PTR-MS instrument (Müller et al., 2014) deployed aboard the NASA P3B aircraft during the California campaign in January and February 2013 and in Colorado in July and August 2014. The P3B flight pattern was specifically designed for satellite validation: the aircraft flew repeated upward and downward spirals, typically 5 km wide connected by transects. An example trajectory, overlaid on the locations of CrIS $NH_3$ retrievals, is shown in Fig. 1: the aircraft altitude is indicated by the colors, and the locations of the spirals are marked with letters. Note that the PTR-MS instrument samples the atmosphere at 1 Hz but the data in Fig. 1 were binned over 100 m to improve visibility. The estimated instrument uncertainty is 35 % (Müller et al., 2014). However, the PTR-MS $NH_3$ data were a side product of the PTR-MS measurements during DISCOVER-AQ, which were designed to obtain data on volatile organic compounds (VOCs). $NH_3$ is sticky and accumulates in the instrument inlet, slowing the instrument response. This effect leads to biases if the $NH_3$ amounts are changing rapidly (Sun et al., 2015); when the aircraft is leaving the boundary layer on upward spirals, the instrument does not respond quickly enough to the sharp decrease in $NH_3$ and overestimates the $NH_3$ concentration; similarly, when entering the boundary layer on downward spirals, the response to the increase in $NH_3$ is slow, and $NH_3$ is underestimated (see Fig. 9 in Guo2021). Furthermore, the detection limits for $NH_3$ were much higher than for the VOCs that were the primary target of the PTR-MS measurements: 7.0 ppbv in California and 3.0 ppbv in

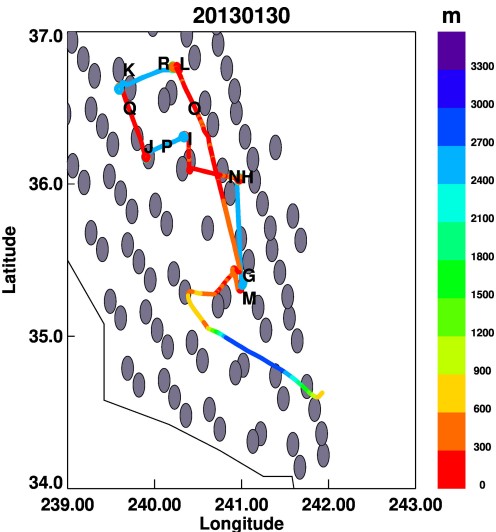

**Figure 1.** Sample aircraft track during DISCOVER-AQ in the Central Valley, CA; colors indicate altitude, letters locations of aircraft spirals; CrIS pixels are shown as gray ellipses.

Colorado (Armin Wisthaler, personal communication, 2023). These limits imply that any aircraft observations below these values are effectively noise.

### 3.2 Ground data

The USDA-ARS Northwest Irrigation and Soils Research Laboratory established a regional $NH_3$ monitoring network in the Magic Valley region of south-central Idaho, USA, utilizing the NADP AMoN (Puchalski et al., 2015) network technology and protocols, but with much greater spatial density. The network measured ambient $NH_3$ concentrations along two transects of the Magic Valley (north–south and west–east) utilizing passive diffusive $NH_3$ samplers collected on a bi-weekly basis from February 2018 through December 2020. The objective of the project was to determine the spatial variability of ambient $NH_3$ concentrations across the region, which is dominated by agricultural production and high-density dairy operations, to better understand the potential for $NH_3$ transport within and downwind of the region.

### 4 DISCOVER-AQ analysis

The DISCOVER-AQ campaigns in California and Colorado provide the most comprehensive set of in situ $NH_3$ profile data available (as opposed to retrievals from FTIR instruments). Both locations have many strong sources and each campaign carried out multiple flight days over a 2-month period. These datasets demonstrate the strengths and limitations of satellite data in areas of great interest to the air quality community; additionally, they allow for the evaluation of the accuracy of the retrieval estimated error, as calculated

from Eq. (3). During each flight the aircraft flew multiple up and down spirals. The satellite profiles were co-located with aircraft profiles taken within 1 h of the satellite overpass time and 15 km of the pixel center, the same criteria used by Guo2021. This co-location criterion is much stricter than is usual for satellite validation (see Hegarty et al., 2022 for an example with CO retrievals from AIRS, which used 9 h and 50 km) but is necessary given the short lifetime of NH$_3$ (on the order of hours to days) due to its high reactivity and fast deposition. In fact, Tournadre et al. (2020) used an even stricter time requirement of 30 min for comparing FTIR and IASI NH3 retrievals over Paris, but we found that such a limited time window drastically reduced the available data. Given the chosen criteria, each CrIS or AIRS profile was compared with data from at most two spirals. Retrievals were checked for quality by ensuring that for all retrievals the root mean square error (RMSE) of the residuals was less than 5.0. The MUSES cloud optical depth (COD) values were also evaluated but since the maximum COD for the retrieved profiles was 0.25, no retrievals were rejected due to large COD. Four CrIS profiles over Colorado were rejected due to very high estimated uncertainties. The DOFs ranged between 0.8 and 1.1, except for two CrIS profiles over California, four AIRS profiles in California and six AIRS profiles in Colorado, for which the DOFS were smaller (0.2–0.7). Given the small number of profiles in each dataset, we did not exclude any profiles based on the DOFs.

Comparing the aircraft and satellite profiles requires regridding the aircraft data to the satellite vertical grid. This was accomplished by defining layers centered around each AIRS/CrIS level, then finding the median value of the aircraft profile in each layer. The lowest layer extended from the surface to the mid-point between the surface and the first AIRS/CrIS level above the surface. The CrIS retrieval layers are fairly coarse and therefore the median value of the PTR-MS is derived from a set of hundreds of measurements spanning the layer, and from both up and down flight paths, thus possibly reducing to some degree the biases from entering and leaving the boundary layer discussed earlier. In the California campaign the average height of the mixed layer was around 500 m and never exceeded 1 km ($\sim$ 900 hPa), and thus these biases were only present in the layer centered around this level, where levels of NH$_3$ (5–40 ppbv) are still high. On the other hand, in Colorado the average height of the boundary layer was around 3 km ($\sim$ 700 hPa), and therefore the biases were present in a layer with lower levels of NH$_3$ (0–5 ppbv). Example plots of the distribution of the aircraft data in each layer (Fig. S4) demonstrate the variability in these data, and show that the median and mean do not always coincide, indicating a non-Gaussian distribution of the measurements in these cases and therefore suggesting the use of the median as the most appropriate metric.

## 4.1 California

DISCOVER-AQ in California took place during January and February 2013. The Central Valley is one of the strongest NH$_3$ source regions in North America (e.g., Clarisse et al., 2009; Shephard et al., 2020), and this was reflected in the aircraft data, which registered near-surface amounts as high as 100 ppbv (Fig. 2a). There were thermal inversions over the entire period (Fig. 2b), which led to increased uncertainties in the retrieval, as they effectively created an emission layer above the surface, i.e., a layer that is warmer than the surface and therefore emits more than it absorbs. Inversions also limit the vertical extent of the boundary layer, with consequently lower NH$_3$ concentrations at altitudes where the retrieval has greater sensitivity. Nevertheless, evaluating the AIRS and CrIS NH$_3$ profiles against the aircraft data is a useful exercise, as the combination of inversions and strong sources is not a rare occurrence, and this analysis demonstrates both the capabilities and limitations of retrievals under these conditions.

A total of 43 AIRS and 58 CrIS profiles met the aforementioned co-location criteria. Before applying the instrument operator as described in Sect. 2.1, we directly compare the satellite data against the aircraft profiles (Fig. 3a and b). This is done to introduce the satellite data, and to demonstrate to users unfamiliar with the instrument operator the importance of accounting for the different vertical resolution of the remote and in situ instruments through the use of the operator. Both instruments showed large negative biases near the surface, as low as $-80$ ppbv for AIRS and $-100$ ppbv for CrIS, while the average bias at this level was $\sim -38$ ppbv for AIRS and $\sim -44$ ppbv for CrIS, with a spread of $\sim 24$ and 38 ppbv, respectively (see Table 1). This large negative bias is likely due to a combination of sub-pixel heterogeneity (Sun et al., 2015; Kille et al., 2019), the inherent difficulties of carrying out retrievals over thermal inversions, and systematic and smoothing errors (due to the different vertical resolution). Note that the mean CrIS value at the surface ($\sim$ 16 ppbv) is 60 % greater than the AIRS value ($\sim$ 10 ppbv), while the reverse is true at 908 hPa (7.6 ppbv vs. 9.7 ppbv). This difference can be attributed to the greater number (24 out of 43) of moderate (green curves) or background (blue curves) a priori profiles selected by the MUSES algorithm for the AIRS retrievals, when compared with CrIS (4 out of 48 TS4). The moderate and background a priori profiles and constraints drive the retrieval to distribute NH$_3$ more uniformly over the vertical range, rather than concentrating it near the surface. For either instrument the surface mean value is much lower than the PTR-MS mean ($\sim$ 37 ppbv), consistent with the large biases noted earlier, although the CrIS mean value ($\sim$ 16 ppbv) is just within the standard deviation of the PTR-MS data. Clouds are accounted for in the MUSES retrieval (Kulawik et al., 2006) and thus cloudy conditions should not significantly impact the NH$_3$ retrievals. Both the bias and the spread drop significantly with increasing alti-

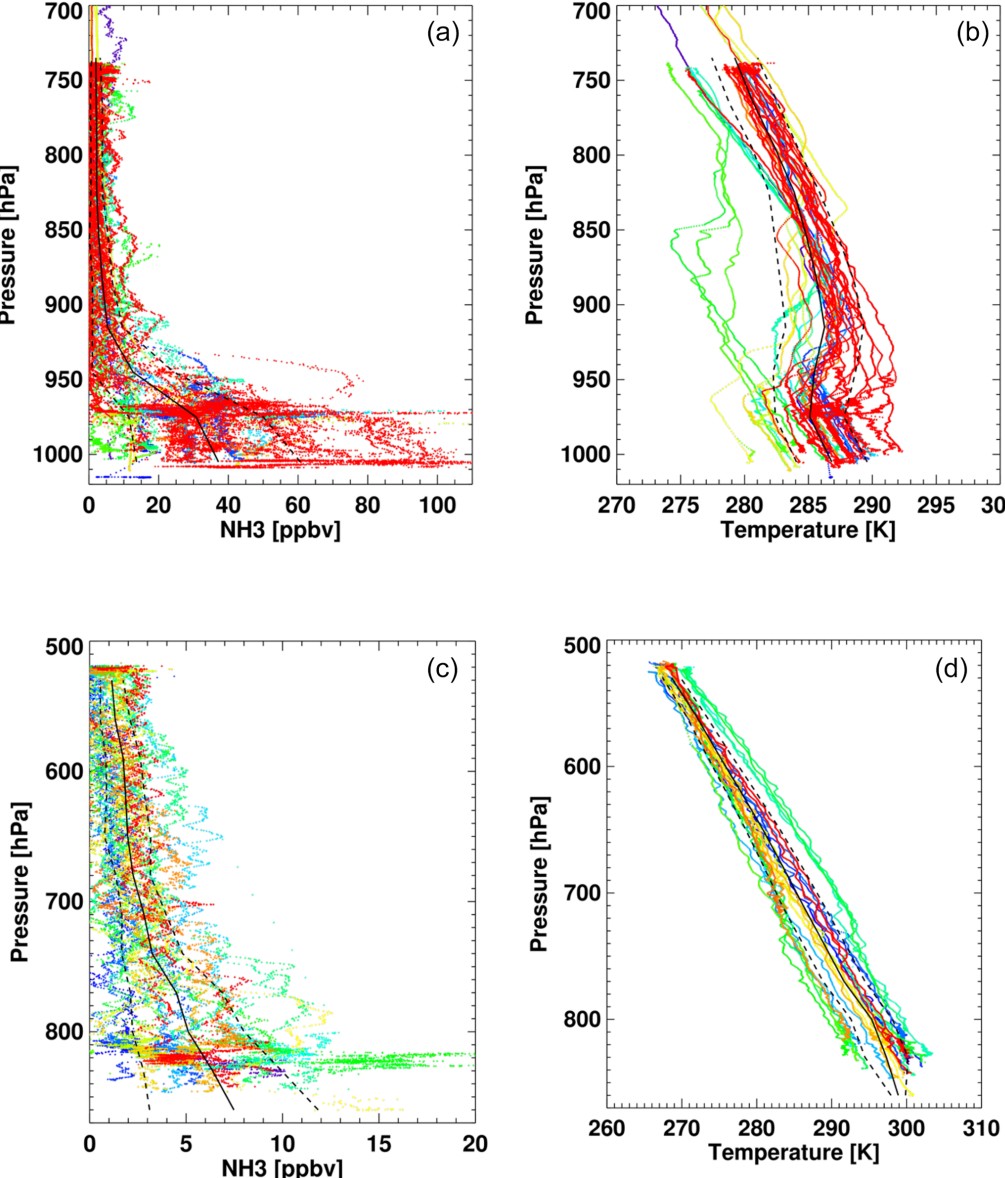

**Figure 2.** $NH_3$ profiles **(a, c)** obtained from aircraft during DISCOVER-AQ in California **(a, b)** and Colorado **(c, d)**; only profiles co-located with CrIS data are shown; corresponding temperature profiles **(b, d)**. Solid black line indicates the mean, dashed black lines the standard deviation.

tude, as do the measured concentrations; while the biases decrease, they become positive and are not insignificant at 825 hPa ($\sim 30\%$ at 825 hPa), and the spread is quite large ($\sim 100\%$). However, at this level, which is above the mixed layer height (MLH) for all flight days, many of the aircraft observations are below the detection limit of the PTR-MS (see Fig. S5) and are thus highly uncertain.

Therefore it become impossible to make any quantitative statements about the differences between the satellite and aircraft data at these higher altitudes.

The sum of the rows of the averaging kernels (SRAK) (Fig. 4), which provides an estimate of the retrieved infor-

mation at each level originating from the measurement rather than from the a priori, shows for both AIRS and CrIS that while the information from the radiance data peaks just below 700 hPa, it also significantly contributes to the retrieved surface values. This is driven by the structure of the co-variance matrix ($S_a$). As noted in the introduction of the DISCOVER-AQ section, the DOFS for AIRS and CRIS NH3 ranged mostly between 0.8 and 1.0, signifying the retrieval provides only one piece of information, basically a column amount. By building off-diagonal correlations in a priori co-variance matrix between the surface level and a few levels above, this information is vertically distributed in such a way

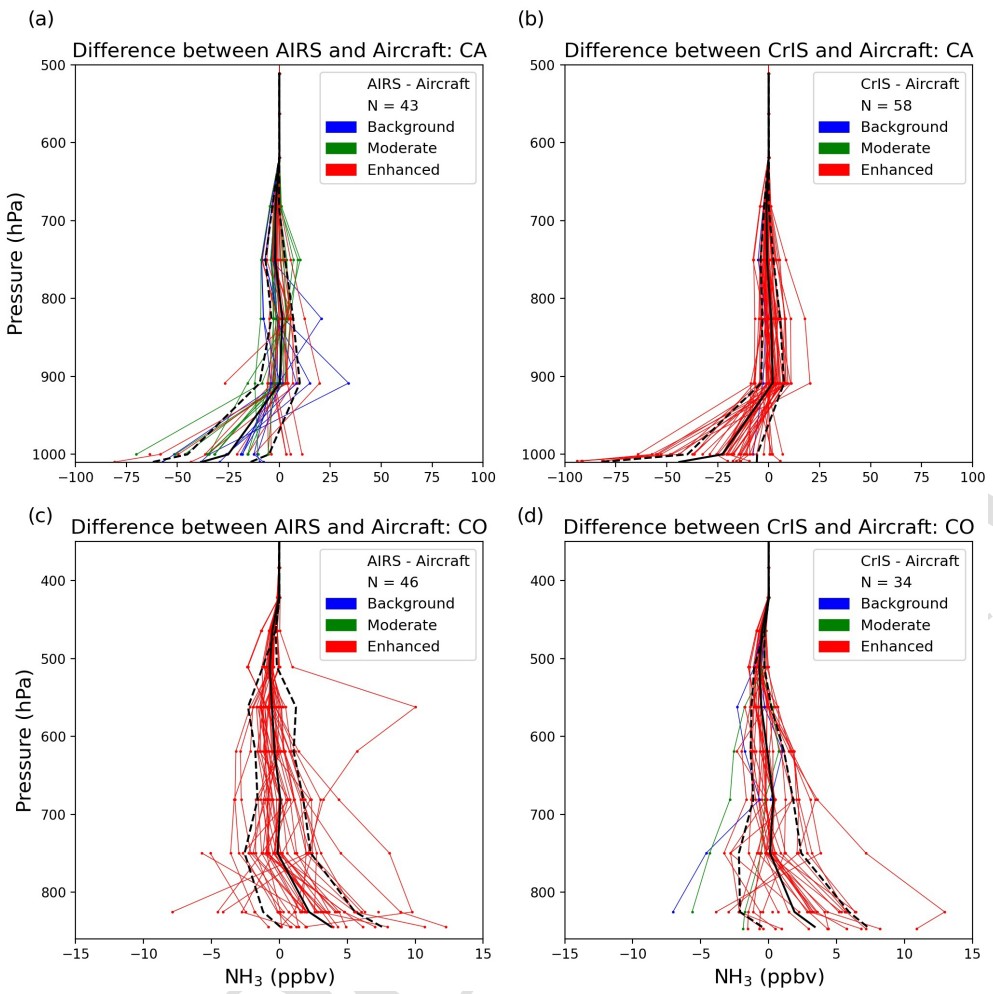

**Figure 3.** Difference between AIRS and aircraft NH$_3$ profiles **(a, c)** and CrIS and aircraft **(b, d)** during DISCOVER-AQ in California **(a, b)** and Colorado **(c, d)**; solid line indicates mean bias and dashed lines standard deviations. Colors indicate choice of a priori profile.

that it restricts unphysical oscillations in the retrieved profile and deviations in the a priori profile shape. Each of the three a priori profiles is associated with a different covariance matrix. The enhanced a priori retrieval tends to load the profiles at the surface, while the moderate and background profiles push NH$_3$ to the free troposphere. The maximum SRAK ($\sim$ 1.2–1.3) is similar for both sensors, as is the mean SRAK at the surface ($\sim$ 0.6). However, there is much greater variability in the AIRS SRAK, possibly due to higher instrument noise.

Applying the instrument operators following Eq. (4) (Fig. 5a and b, red and black curves) eliminates the smoothing error, which reduces the bias at all levels to close to or less than 1.0 ppbv (Table 1, top section), TS5 roughly 7 %– 10 % of the surface value, increasing to 20 % at 825 hPa, indicating that the large differences seen in the direct comparison are due mainly to the inability of the satellite instruments to resolve the fine vertical structure of the profile. The standard deviation is also reduced, especially at the lower levels,

but is not eliminated, which suggests that other error sources are present. If instead the a priori profiles are compared with the aircraft data, we see a large negative bias near the surface ($\sim$ −9 ppbv for AIRS and $\sim$ −18 ppbv for CrIS). This result demonstrates that the retrieval process adds significant information and reduces the a priori error.

A great benefit of the optimal estimation approach is that is provides both retrieved profiles and estimated errors. If these errors are lower than the error relative to the in situ or model data, they indicate that some error sources have not been accounted for in the retrieval. Here we compare the sum of the measurement and systematic errors from Eq. (3) (Fig. 6a and b, red and blue curves) against the measured uncertainties (the fractional standard deviation derived from the standard deviations of the differences in the "Satellite-Aircraft with AK" column in Table 1) (Fig. 6a and b, black curve), which have the smoothing error removed by the application of the averaging kernel, **A**. Since the measurement error is usually quite low (see Fig. S3 for examples) the estimated

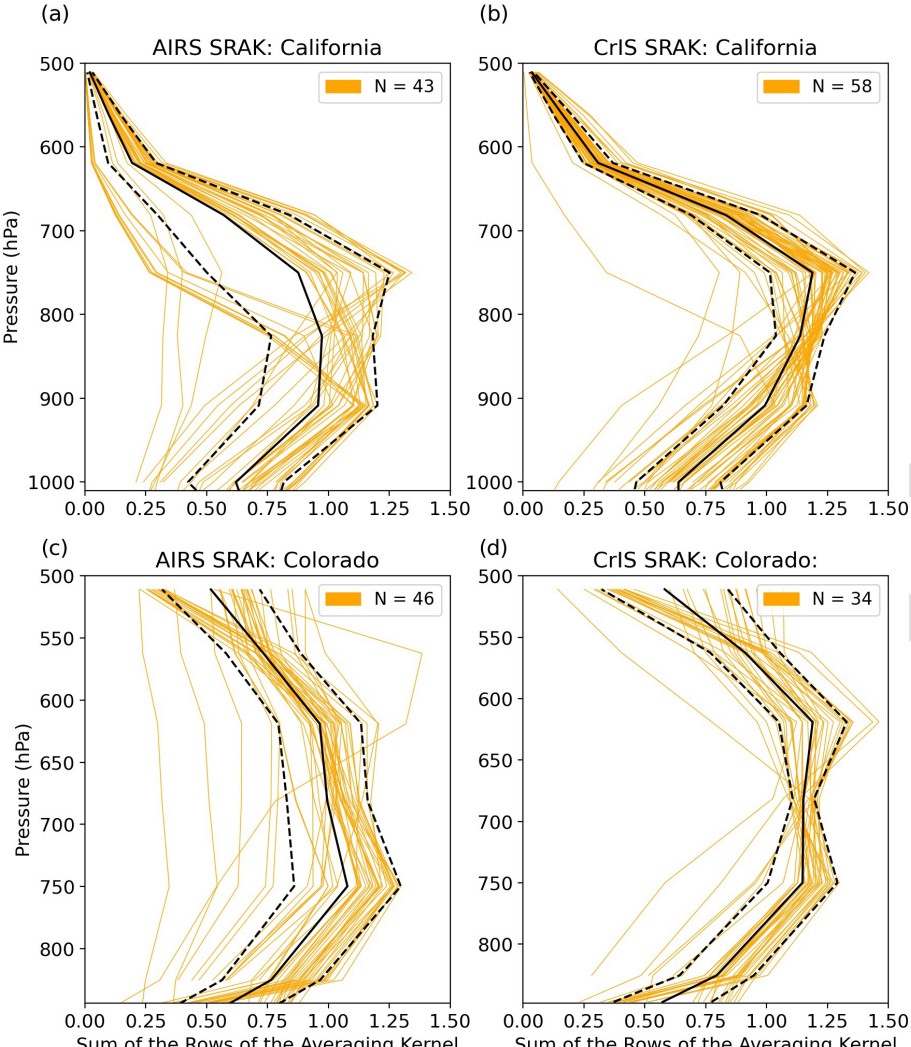

**Figure 4.** Sum of the row of the averaging kernels (SRAK) for AIRS **(a, c)** and CrIS **(b, d)**; California **(a, b)** and Colorado **(c, d)**.

uncertainties are nearly equivalent to the systematic errors, and thus represent the estimate of the error in $NH_3$ due to errors in temperature and water vapor. The average estimated uncertainties are below the measured uncertainties for both 5 AIRS and CrIS, confirming that some error sources are not accounted for in the optimal estimation process.

A possible candidate is the sampling difference: AIRS and CrIS sampled three-dimensional columns, 15–50 km wide, while aircraft instruments sampled in two dimensions, ver- 10 tically and along a spiral line with a 5 km width. Sub-pixel variability likely also contributes to these errors. The estimated uncertainties range in general from 5 % to 50 % at the surface, although five cases had significantly larger errors. At 850 hPa the variability in the uncertainties is much larger, 15 ranging from 10 % to 250 %, strengthening the argument that it is not possible to carry out a meaningful comparison at this altitude or higher with this dataset.

Much of the work on validating $NH_3$ from space-based infrared sensors has been done using IASI data, which provide total columns rather than profiles (e.g., van Damme 20 et al., 2015a; Dammers et al., 2017; Guo2021), although Dammers et al. (2017) estimated IASI profiles by using two fixed vertical profiles to convert column amounts to profiles and thus to also obtain surface values. Guo2021 explored four approaches (see Fig. 5 in Guo2021) to account for the 25 $NH_3$ amount above the mixed layer height (MLH) and obtained the best agreement with IASI data by assuming zero $NH_3$ above the MLH ($R = 0.57$ and slope $= 1.0$). This is a fairly reasonable assumption, since $NH_3$ has a short lifetime (on the order of hours or days), and is rarely trans- 30 ported to the middle or upper troposphere, except during strong fires; it has been measured above the mixed layer, but at low levels (less than 1 ppbv; e.g., Höpfner et al., 2019; Nowak et al., 2010). Here we compare AIRS and CrIS total columns with aircraft total columns calculated following 35

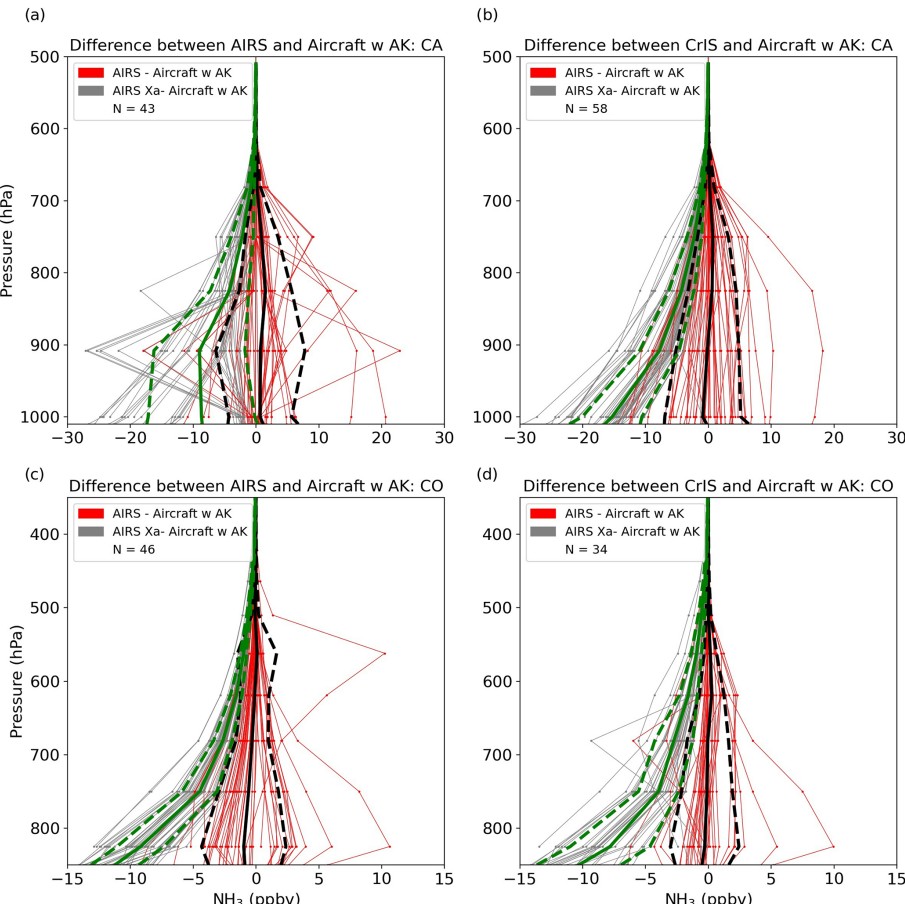

**Figure 5.** Difference between AIRS and aircraft NH$_3$ profiles **(a, c)** and CrIS and aircraft **(b, d)** during DISCOVER-AQ in California **(a, b)** and Colorado **(c, d)**. The averaging kernel has been applied to the aircraft data; red lines show differences between the retrieved profiles and the aircraft data, solid black lines indicate mean bias and dashed black lines standard deviations; gray lines show differences between the a priori profiles and the aircraft, solid green lines indicate mean bias and dashed green lines standard deviations.

two approaches. First we used the zero NH$_3$ above the MLH approach adopted by Guo2021. In this procedure the in situ NH$_3$ total column is estimated by integrating the aircraft NH$_3$ profile to the top of the mixed layer; above this level, NH$_3$ amounts are assumed to be zero.

The MUSES total columns are compared against the integrated aircraft columns, using orthogonal linear regression (Fig. 7a and b), again colored by the choice of the a priori profile; the intercept has been allowed to vary, as both AIRS and CrIS have detection limits, ($\sim 1.0$ ppbv, for thermal contrast above 5 K), as does IASI ($3.0 \times 10^{15}$ molec. cm$^{-2}$, for thermal contrast above 5 K). Note that in this approach the instrument operators have not been applied to the aircraft column data. The correlation coefficients are 0.58 for AIRS and 0.62 for CrIS, within the range of results from previous studies (e.g., Dammers et al., 2017), found correlations ranging from 0.28 to 0.81 when comparing surface FTIR and CrIS columns. However, the slopes are very much greater than 1 (1.9 for AIRS and 1.6 for CrIS), which was not the case in the IASI evaluation of Guo2021. If this assumption of zero

amounts of NH$_3$ above the mixed layer is no longer valid at the later (13:30 LST) CrIS and AIRS overpass times, the aircraft columns would be biased low with respect to CrIS and AIRS; the profile differences shown in Table 1 also suggest that AIRS and CrIS are measuring more NH$_3$ above 910 hPa. Given the uncertainty in the aircraft measurements at these higher altitudes, it is not possible to currently determine the true NH$_3$ amounts above the mixed layer.

In the second approach (Fig. 8) we calculate partial AIRS/CrIS columns by integrating the profiles to the MLH only. The results are interesting: the AIRS correlation (Fig. 8a) decreases (from 0.58 to 0.46), and the slope drops from 1.9 to 1.4. These changes occur because many of the columns decrease substantially (note large group of values below $1.0 \times 10^{16}$ molec. cm$^{-2}$). We found that these observations derive from profiles for which the retrieval selected a background or moderate prior. When a background or moderate prior is chosen, the retrieval distributes the NH$_3$ differently than if the enhanced prior is used (see Fig. S3, two rightmost panels): in the moderate case the retrieval creates a

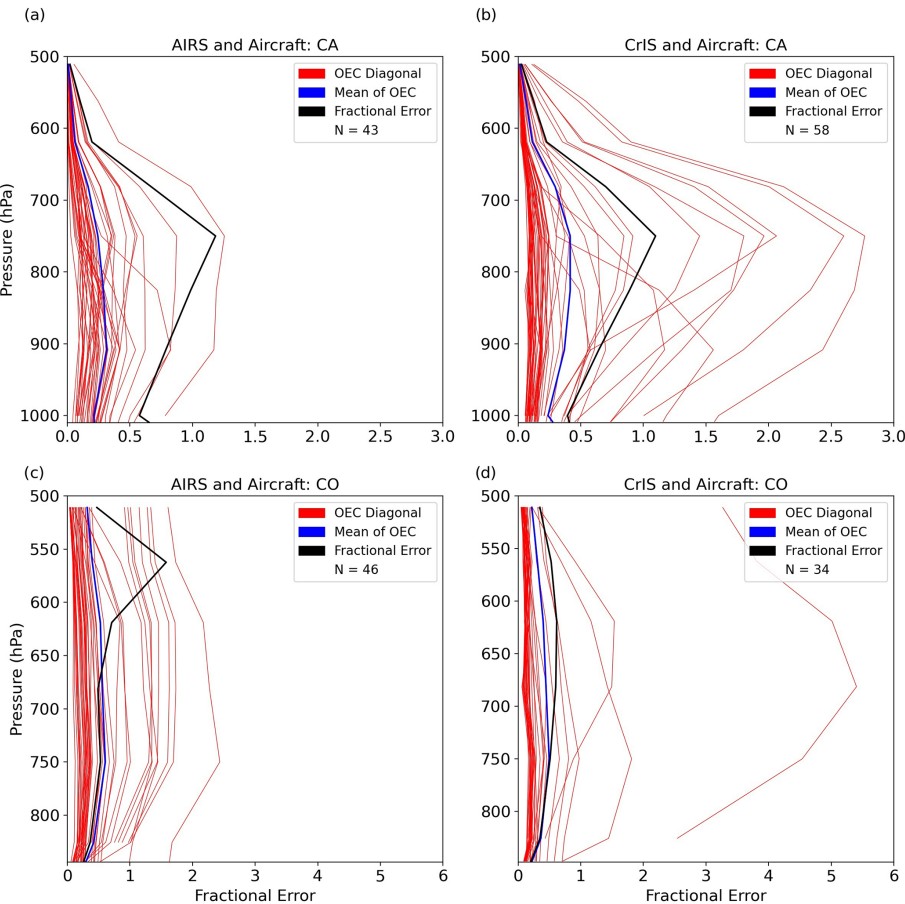

**Figure 6.** Fractional standard deviation between AIRS **(a, c)** and CrIS **(b, d)** and aircraft profiles with the averaging kernel applied (black) during DISCOVER-AQ in California **(a, b)** and Colorado **(c, d)**; estimated uncertainty (red) and mean of estimated uncertainty (blue).

profile that peaks around 900 hPa, while in the polluted case the peak is at the surface. By limiting the integration to the top of the MLH, a substantial fraction of the retrieved column from the AIRS observations is excluded for the profiles derived from the moderate prior. The slope of the CrIS comparison drops dramatically from 1.6 to 0.6, but the correlation actually increases slightly from 0.61 to 0.64. Visual inspection suggests that truncating the CrIS integration had a large impact on a few outliers but overall reduced most columns only slightly, since the bulk of the NH$_3$ is below the MLH, and thus did not substantially affect the correlation. Both the AIRS and CrIS results demonstrate that the retrievals are showing a non-negligible amount of NH$_3$ above the mixed layer, more so for AIRS than for CrIS, but at present it is not possible to determine whether these values are real, given the uncertainties in the aircraft data at these altitudes, or whether they are the result of redistribution of NH$_3$ to higher altitudes driven by the a priori profile shape and error covariance.

### 4.2 Colorado

DISCOVER-AQ Colorado took place during July and August 2014, in the Colorado Front Range. While this is also a region with strong NH$_3$ sources, the aircraft data showed lower values than in the California Central Valley (Fig. 2c); maximum values are on the order of 20 ppbv, with most near-surface values ranging from 5 to 10 ppbv, above the uncertainty in the aircraft data, which was $\sim$ 3ppbv in the Colorado campaign. In contrast to the Central Valley there were no thermal inversions during this period (Fig. 2d) and the MLH was much higher, about 3 km ($\sim$ 700 hPa) on average. Applying the co-location criteria yielded 46 AIRS profiles but only 34 CrIS profiles, in part due to some poor quality CrIS retrievals. Direct comparisons of the AIRS and CrIS profiles with aircraft data (Fig. 3c and d) show both AIRS and CrIS biased high by 3.8 ppbv (AIRS) and 3.4 ppbv (CrIS) near the surface ($\sim$ 844 hPa) and by 2.2 ppbv (AIRS) and 1.9 ppbv (CrIS) at 825 hPa, but the bias is close to zero at 750 hPa. Note that this is in direct contrast to the California results, which showed both AIRS and CrIS NH$_3$ biased very low at and near the surface and biased slightly high at greater al-

**Table 1.** Statistical analysis of the DISCOVER-AQ data from AIRS, CrIS and the aircraft over California in 2013 and Colorado in 2014.

California

| | Profile | | | | Satellite–Aircraft: no AK | | | | Satellite–Aircraft: with AK | | | |
|---|---|---|---|---|---|---|---|---|---|---|---|---|
| | Mean | | SD | | Bias | | SD | | Bias | | SD | |
| Pressure hPa | AIRS ppbv | CrIS ppbv | AIRS ppbv | CrIS ppbv | AIRS ppbv | CrIS ppbv | AIRS ppbv | CrIS ppbv | AIRS ppbv | CrIS ppbv | AIRS ppbv | CrIS ppbv |
| 1008.486 | 9.69 | 16.211 | 10.785 | 6.027 | −38.03 | −43.799 | 23.972 | 37.988 | 1.124 | −0.279 | 5.6 | 6.723 |
| 1000 | 9.342 | 14.443 | 9.875 | 5.44 | −25.049 | −22.746 | 19.995 | 17.106 | 0.664 | −0.89 | 5 | 6.046 |
| 908.514 | 9.693 | 7.63 | 8.634 | 4.101 | 0.238 | 1.956 | 9.81 | 5.582 | 0.691 | −0.159 | 7.08 | 5.015 |
| 825.402 | 5.742 | 4.883 | 5.401 | 3.372 | 1.471 | 1.266 | 5.327 | 4.345 | 1.511 | 0.593 | 4.169 | 3.815 |
| 749.893 | 3.083 | 2.958 | 3.47 | 2.311 | −1.96 | −0.801 | 4.815 | 3.101 | 0.883 | 0.729 | 2.605 | 2.446 |
| 681 | 0.979 | 1.195 | 0.814 | 0.664 | −1.78 | −1.083 | 1.597 | 1.111 | 0.156 | 0.21 | 0.551 | 0.689 |
| 618.966 | 0.244 | 0.357 | 0.149 | 0.083 | 0.046 | 0.038 | 0.057 | 0.076 | 0.008 | 0.014 | 0.046 | 0.078 |

Colorado

| | Profile | | | | Satellite–Aircraft: no AK | | | | Satellite–Aircraft: with AK | | | |
|---|---|---|---|---|---|---|---|---|---|---|---|---|
| | Mean | | SD | | Bias | | SD | | Bias | | SD | |
| Pressure hPa | AIRS ppbv | CrIS ppbv | AIRS ppbv | CrIS ppbv | AIRS ppbv | CrIS ppbv | AIRS ppbv | CrIS ppbv | AIRS ppbv | CrIS ppbv | AIRS ppbv | CrIS ppbv |
| 844.469 | 10.341 | 9.805 | 3.415 | 3.801 | 3.817 | 3.387 | 3.741 | 3.871 | −0.89 | −0.5026 | 2.884 | 2.133 |
| 825.402 | 8.214 | 7.529 | 3.064 | 3.595 | 2.19 | 1.902 | 3.377 | 4.006 | −0.978 | −0.292 | 3.355 | 2.723 |
| 749.893 | 3.889 | 3.852 | 2.173 | 2.101 | −0.12 | 0.102 | 2.431 | 2.292 | −0.592 | −0.108 | 2.375 | 2.021 |
| 681.291 | 2.324 | 2.703 | 1.143 | 1.207 | 0.084 | 0.338 | 1.684 | 1.465 | −0.315 | −0.046 | 1.29 | 1.652 |
| 618.966 | 1.474 | 1.858 | 0.943 | 0.974 | −0.356 | −0.102 | 1.41 | 1.228 | −0.099 | 0.262 | 1.115 | 0.983 |
| 562.342 | 1.07 | 1.104 | 1.444 | 0.548 | −0.538 | −0.535 | 1.764 | 0.724 | 0.084 | 0.159 | 1.557 | 0.498 |
| 510.898 | 0.521 | 0.552 | 0.231 | 0.212 | −0.695 | −0.687 | 0.523 | 0.374 | −0.068 | 0.008 | 0.278 | 0.189 |

titudes. However, here too the mean AIRS (10.3 ppbv) and CrIS (9.8) surface values are within the standard deviation of the aircraft data.

The sum of the rows of the averaging kernel plots (Fig. 4c and d) present similar maximum and surface values as in California, and again applying the instrument operator reduced the bias to below 1.0 ppbv ($\sim 10\%$), but did not reduce the spread in the satellite–aircraft differences dramatically (Fig. 5a and b), while the large bias at the surface seen against the a priori profiles ($\sim -12$ ppbv for AIRS and $\sim -10$ ppbv for CrIS) is removed by the retrieval, confirming again that the retrieval process provides information content beyond the a priori.

The uncertainty analysis (Fig. 6a and b) shows that in the region where both the satellite and aircraft observations are reasonably robust, the estimated uncertainty is quite close to the measured uncertainty below 700 hPa. Finally, the total column comparisons also indicate that the AIRS and CrIS $NH_3$ (Fig. 7c and d) columns are correlated with the aircraft values (with correlation coefficients of 0.56 for AIRS and 0.42 for CrIS), but are biased high, most notably for CrIS, for which the regression presents a slope of 2.3. The AIRS total columns (Fig. 7c) and partial columns (Fig. 8c) are very similar, indicating the most of the retrieved $NH_3$

is below the MLH; since all AIRS retrievals used the enhanced prior, which concentrates $NH_3$ in the surface layer, this is to be expected. The CrIS partial columns (Fig. 8d) show marked reductions in the smaller column values, which increase the slope from TS6 2.4 to 3.5, and reduce the correlation to 0.31. These smaller columns correspond to retrievals that used background or moderate a priori profiles, and, as was described in the California section, push $NH_3$ higher up in the column, and are thus more sensitive to truncation. The fact the columns from the satellite instruments are much higher than the aircraft columns is possibly due to $NH_3$ produced by fires and lofted above the MLH, which were not measured by the PTR-MS but would be detected by AIRS/CrIS and redistributed downward by the retrieval, leading to the high biases seen in the surface values before applying the instrument operator.

## 5 Magic Valley analysis

### 5.1 USDA network

The USDA-ARS established and maintained an $NH_3$ monitoring network along two transects (north–south and west–

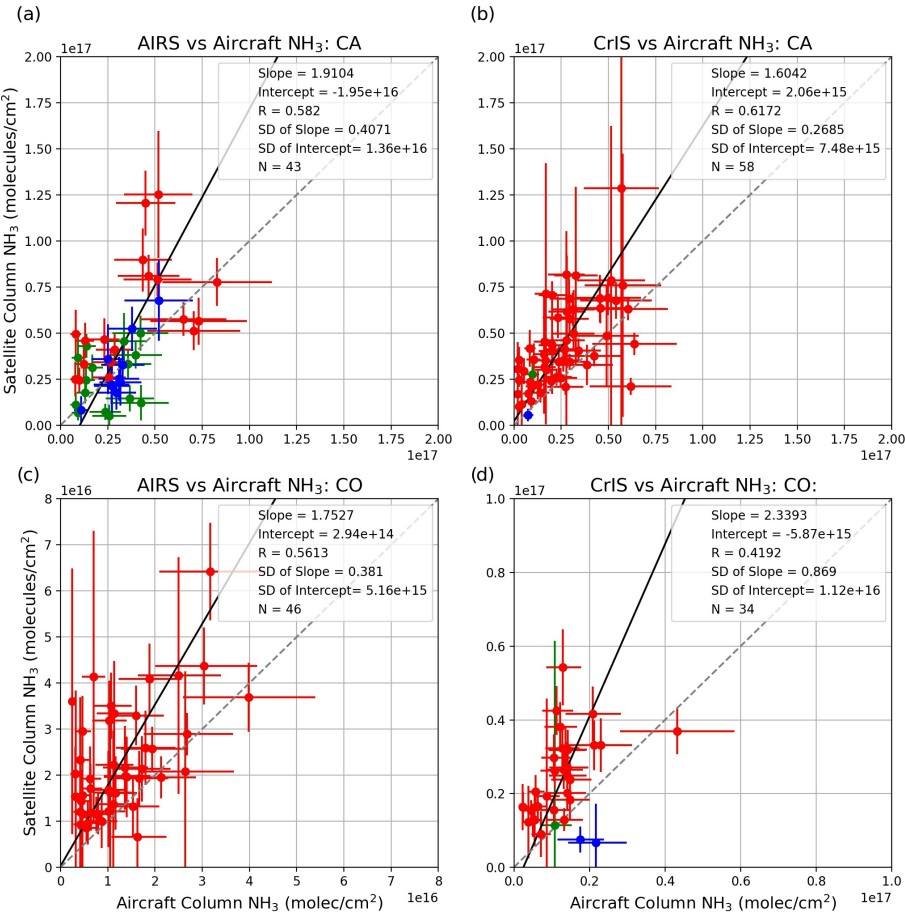

**Figure 7.** Total NH$_3$ columns from AIRS **(a, c)** and CrIS **(b, d)** versus aircraft columns during DISCOVER-AQ in California **(a, b)** and Colorado **(c, d)**; dashed line shows the 1 : 1 line, solid black line the linear fit; vertical and horizontal lines.

east) across the Magic Valley region of south-central Idaho during the period February 2018–December 2020 (Fig. 9). The Magic Valley region is heavily dominated by irrigated agriculture and is one of the most concentrated dairy production regions in the United States. Research in this region has reported that NH$_3$ emissions from agricultural and dairy production contribute approximately 44 000 MT N yr$^{-1}$ to the atmosphere (Leytem et al., 2021) and that NH$_3$ emissions fluctuate by season following trends in temperature (Leytem et al., 2011, 2013). The network was established to gain a better understanding of the spatial variability of ambient NH$_3$ concentrations and transport within the region. The network consisted of eight sampling locations (seven until 2020, when an additional site was added), and also utilized data from the NADP AMoN site located to the north of the region at Craters of the Moon National Monument. NH$_3$ concentrations were measured with passive diffusive NH$_3$ samplers (Radiello), which were deployed bi-weekly, and generated 2-week mean surface NH$_3$ concentrations. Radiello samplers have been shown to be approximately 9 % biased low (Puchalski et al., 2015). These data provided a unique

opportunity to evaluate the seasonal signals measured by CrIS NH$_3$, as well as its capability to capture small-scale (on the order of a few kilometers) spatial variability.

## 5.2 Evaluating CrIS NH$_3$ against the surface data

CrIS SNPP NH$_3$ data are available for most of the 2018–2020 period, with a gap in the spring of 2019 due to an instrument malfunction. Surface CrIS data within 15 km of each site were compared with the ground data at that site. Profiles with RMSE greater than 5.0 were excluded from the time series, as were clouds with COD greater than 1.0. Only the CrIS observations at $\sim$ 13:30 LST were analyzed. The NH$_3$ retrievals at TS7 13:30 LST have weaker signals (due to lower thermal contrast), and would add uncertainty to the results. While there are strong diurnal cycles in the NH$_3$ emitted from the dairy facilities (Leytem et al., 2011, 2013) the average daily emissions and temperatures, which strongly control the emissions, are close to the early afternoon values. Ideally one would use measurements of the daily cycle in NH$_3$ concentrations, to estimate the ratio between the TS8 13:30 LST and 24 h mean concentrations, as was done by

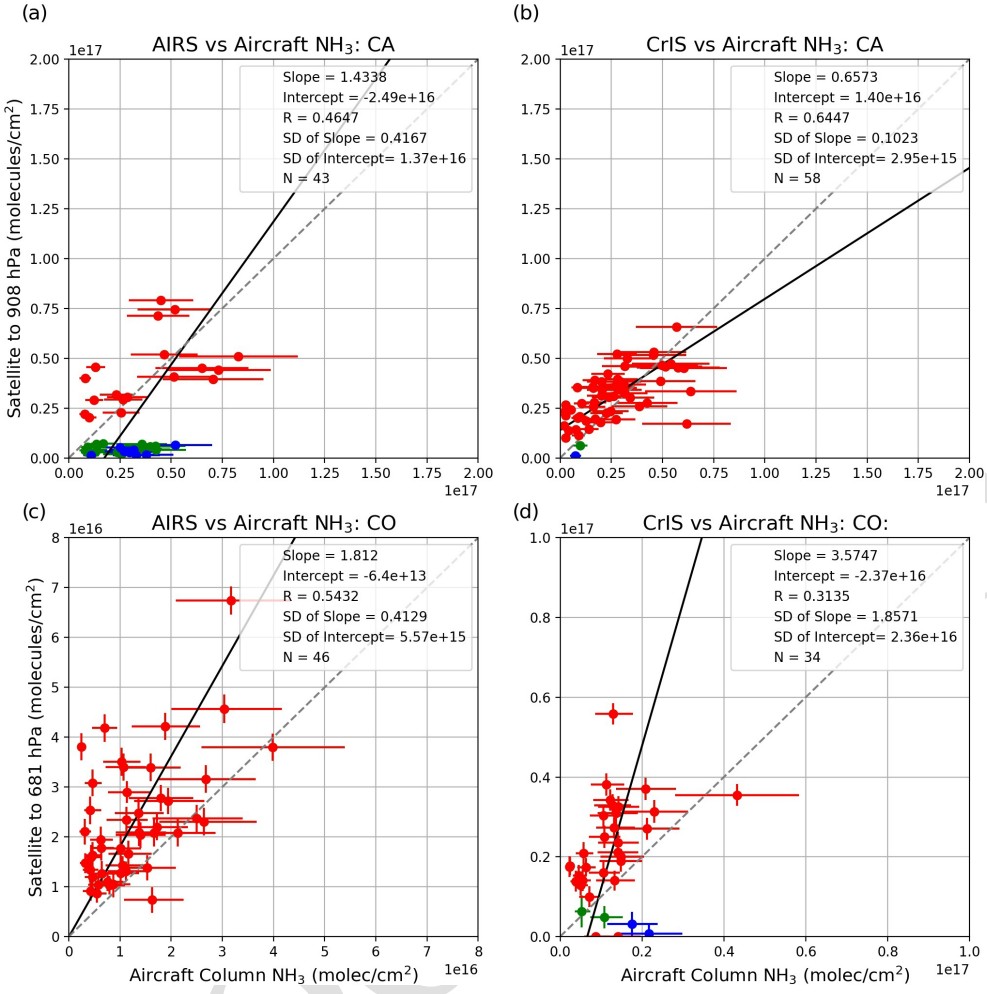

**Figure 8.** Same as Fig. 7, but AIRS/CrIS columns are partial columns, extending only to the top of the MLH.

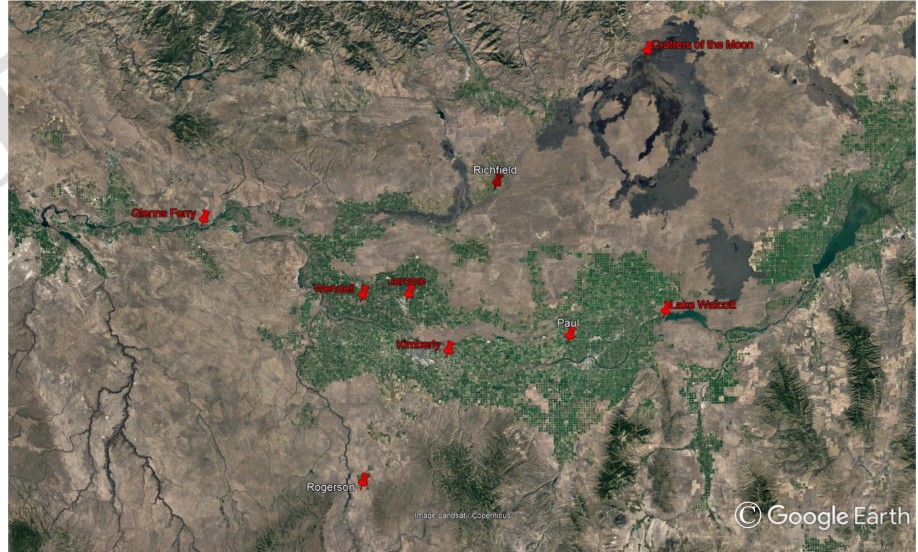

**Figure 9.** USDA Magic Valley network showing the nine measurement sites.

Pinder et al. (2011), but such data are not available for the Magic Valley site. Figure 10 shows the individual CrIS observations (small red triangles), the 2-week averages of the CrIS data (large red triangles connected with a red line) and the ground data (blue triangles) at each site.

The time series in Fig. 10 are sorted by the peak values of the ground data. At every site, CrIS clearly captures the seasonal cycle, although winter values are usually underestimated: this can be attributed to weak radiative signals due to low temperatures and low thermal contrast. The CrIS level of detectability is normally cited as $\sim 1.0$ ppbv (Shephard and Cady-Pereira, 2015), but at low thermal contrast this level increases significantly. A few strong warm season peaks in the ground data are also not captured. At Craters of the Moon, a national monument, and at Rogerson, in land administered by the Bureau of Land Management, CrIS returns consistently higher values than the ground site during the warmer months (May to October). At present we have no definitive explanation for this high bias. At the next four sites (Glenns Ferry, Richfield, Lake Walcott and Kimberly), which are in areas of mixed agricultural activity, CrIS and the ground data are in good agreement during the warmer months, although CrIS underestimates the June 2018 maximum at the Kimberly site. At all four sites there is a peak in the ground data in November 2019 that is either matched in the CrIS data (Glenns Ferry) or at least visible (Richfield, Kimberly, Lake Walcott). This peak is also evident in the data from Paul and possibly Wendell, although the variability at this latter site makes it difficult to confirm. This suggests an area-wide change in meteorological conditions, such as an inversion, that led to increased $NH_3$ near the surface: however, there is no evidence in the meteorological data for such an inversion. The last three sites (Paul, Jerome and Wendell) are close to and/or downwind of multiple dairies, which likely leads to greater sub-pixel inhomogeneity. CrIS underestimates the warm season maximum, and does not capture many of the peaks in the ground data, most notably at the Wendell site, where CrIS did not observe the peaks over 80 ppbv in 2018 and 2019.

Analyzing these data in aggregate, first spatially then temporally, provides some useful insights. Plotting the CrIS 2-week averages against all the ground data values (Fig. 11), and excluding CrIS data with cloud OD greater than 2.0, along with the data from the Wendell site, which are extreme outliers, shows a correlation of 0.6, at the high end of the values reported in the literature (e.g., see van Damme et al., 2015b, as noted in Introduction) and a slope of TS9 0.65, indicating that CrIS $NH_3$ is biased low overall. This result is in line with the low biases found in the surface values of the AIRS and CrIS California data, although here not necessarily caused by thermal inversions, and with the low biases in the CFPR results at high $NH_3$ FTIR values seen by Dammers et al. (2017). It provides a quantitative measure of the agreement seen in the seasonal cycles shown in Fig. 10.

Three years of fairly dense data over a region with many sources with fixed locations are an excellent candidate for spatial oversampling algorithms, which trade temporal resolution for greater information on spatial variability. Here we applied the physics-based oversampling algorithm developed by Sun et al. (2018), which uses the instrument spatial response function to weight the contributions of each satellite observation to a fine grid (here 0.002°), to each of the 3 years of CrIS data taken over the Magic Valley region (Fig. 12). The in situ data are overlaid on the CrIS maps; note that the Wendell values are shown in the upper-right corner of the maps, as otherwise they would distort the color scale; note also that the Jerome data are blank in 2018 and 2019, when this site was not operational.

The location of the CrIS $NH_3$ "hotspots" and the gradients in $NH_3$ are very consistent from year to year, although $NH_3$ concentrations are lower in 2019, possibly due to the CrIS data gap between March and June. There is good qualitative agreement with the in situ data, with the exception of the Wendell site, which was discussed earlier. Moreover, the hotspots are very well correlated with the areas of high dairy density (Fig. 11, upper right). These maps illustrate the power of the CrIS data to provide context to in situ measurements and far more information on the spatial variability in $NH_3$ than is normally available from emissions databases in the United States, which are frequently at county level. Data from these gridded maps could be used to constrain emission inventories over much larger areas and at more frequent intervals than is currently possible. This oversampling analysis demonstrates the utility of providing users with Level 2 products, since they then choose the sampling periods and resolution that are most appropriate for their purposes and are best suited to the times and regions under investigation. For example, a user might want to study the variability of $NH_3$ over only the urban area of Mexico City (e.g., Herrera et al., 2022). Level 3 products provide no such flexibility.

## 6 Conclusions and future work

The comparison between the DISCOVER-AQ aircraft datasets and the co-located AIRS and CrIS data provide useful information for end-users who would like to use CrIS and AIRS data over strong-source regions. Given the large uncertainties in the aircraft data above the MLH, the profiles can only be evaluated within the mixed layer. Average biases in this layer, after smoothing errors are accounted for, are below or close to 1 ppbv. The AIRS and CrIS profiles individually have large estimated uncertainties, ranging from 5 % to 50 %. On average in California, as the error analysis in Fig. 6a and b indicate, the a posteriori estimated error underestimates the actual uncertainties, probably due to the thermal inversions and high sub-pixel variability, which is expected, since these two factors are not accounted for in the error estimate. Ongoing work is attempting to quantify the

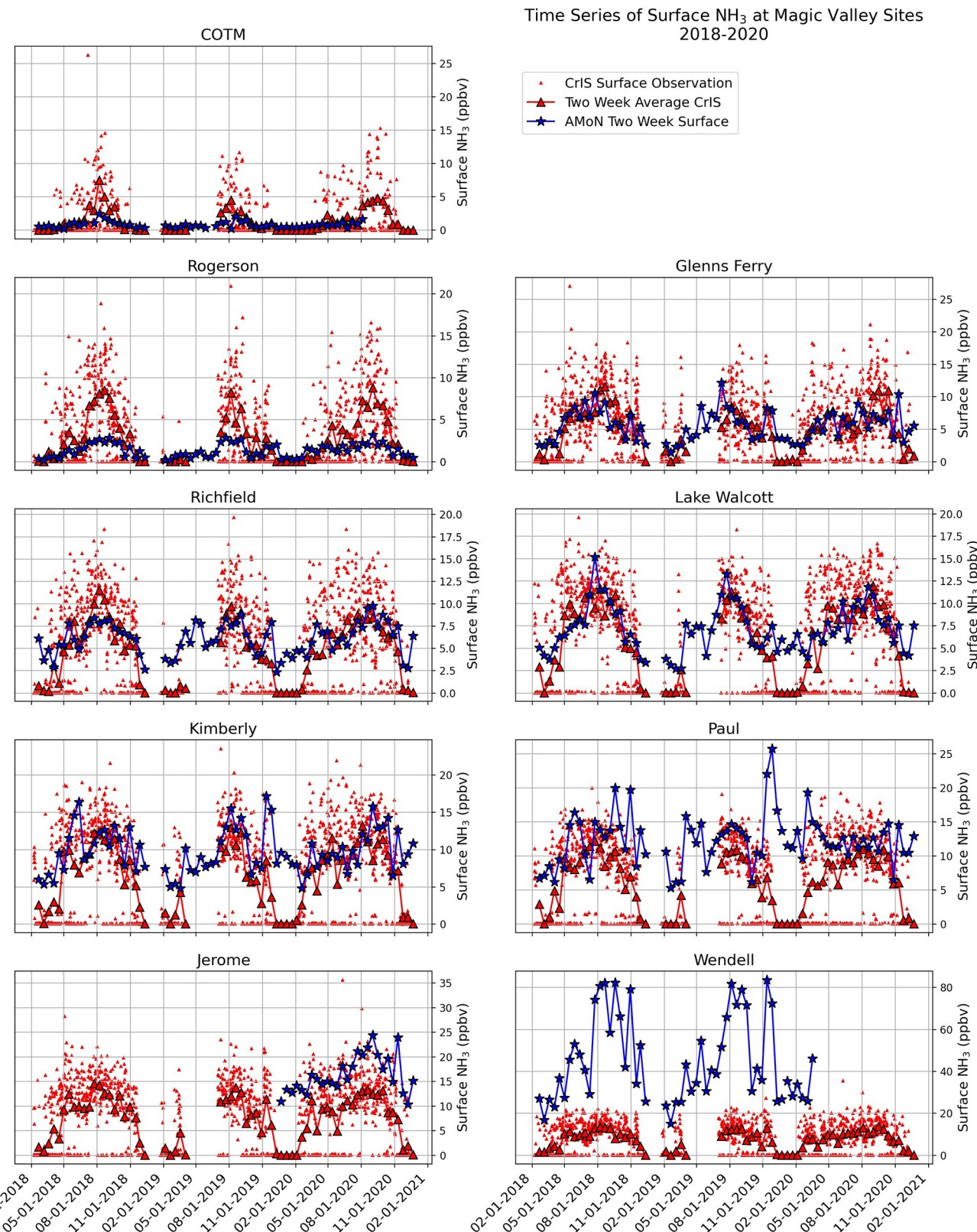

**Figure 10.** Time series of the in situ data (blue triangles) and the collocated CrIS surface values; red dots indicate daily values, red triangles 2-week means.

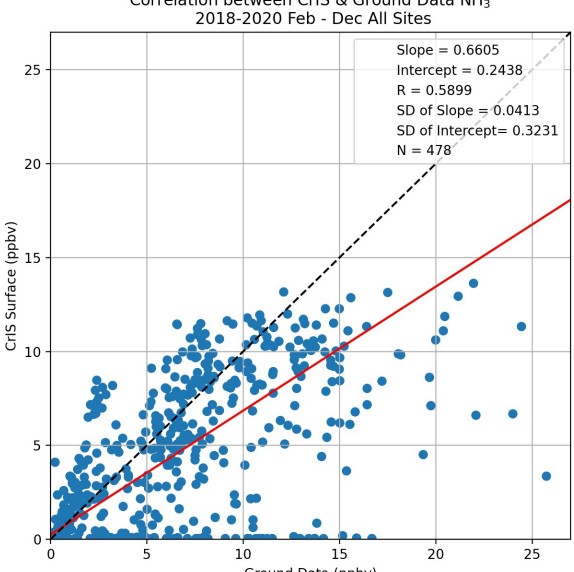

**Figure 11.** Two-week means of surface $NH_3$ from CrIS and the in situ instruments; dashed line shows the $1:1$ line, red line the linear fit.

effect of sub-pixel variability by analyzing data from multiple aircraft passes over the same CrIS pixel; these data were obtained during the TRANSAM (http://catalog.eol.ucar.edu/trans2am, last access: TS10 2022) campaign in 2022. Further data from the HYTES (https://hytes.jpl.nasa.gov/, last access: TS11 2019) instrument flown over the Imperial Valley in California are also being evaluated. Over Colorado the estimated uncertainty is very close to measured uncertainty within the mixed layer, suggesting the error sources are properly accounted for in this region.

This study has not attempted to untangle the impact of the errors in the retrieved water vapor from those of temperature on the $NH_3$ errors. Such an analysis is an important and ongoing task, as global maps of CrIS $NH_3$ have revealed artificial hotspots of $NH_3$ over tropical oceans, where humidity is high. There is a weak water vapor line in the spectral region used in the NH3 retrievals, which is possibly leading to these artifacts.

The column data analysis suggests that either there is nonnegligible $NH_3$ above the top of the aircraft profile, or the retrievals are overestimating $NH_3$ above this altitude, because the a priori profile shape is very different from the true profile shape. More in situ measurements of $NH_3$ in this altitude region, with increased accuracy, and improved $NH_3$ retrievals with different a priori profile shapes (e.g., longer tails and a faster vertical decay), are required to resolve this issue.

The Magic Valley analysis clearly demonstrates the importance of having more than a few dozen data point measurements to obtain useful information from space-based retrievals of $NH_3$. With 464 observations over 3 years, over a limited region, it was possible to obtain a clear picture of the source distribution in the Magic Valley through the application of a physics-based oversampling algorithm. Further work will apply this approach to other regions and times and use the resulting maps to estimate emissions and to improve reactive nitrogen deposition estimates.

*Data availability.* MUSES AIRS and CrIS $NH_3$ products from S-NPP are available via the GES-DISC from the NASA Tropospheric Ozone and Precursors from Earth System Sounding (TROPESS) project at https://doi.org/10.5067/EYXLPVGTSWFF (Bowman, 2021a) and https://doi.org/10.5067/B4TF7ND8A3O7 (Bowman, 2021b) respectively. The AIRS and CrIS datasets matched with aircraft used here for validation are available from Zenodo at https://doi.org/10.5281/zenodo.10038045 (Cady-Pereira, 2023).

*Supplement.* The supplement related to this article is available online at: https://doi.org/10.5194/amt-16-1-2023-supplement.

*Author contributions.* KECP designed the project, carried out the MUSES retrievals, created the preliminary plots, analyzed the results and wrote the manuscript. XG, RW and MZ provided the co-located aircraft data and made suggestions for the manuscript. AL provided the Magic Valley data and contributed relevant text. KS provided the oversampling code and revised the manuscript. VHP and MS provided useful insights and revised the manuscript. CC and EB created all the plots. MM and AW obtained the DISCOVER-AQ data. VK designed and built the MUSES software.

*Competing interests.* The contact author has declared that none of the authors has any competing interests.

ther geographical representation in this paper. While Copernicus Publications makes every effort to include appropriate place names, the final responsibility lies with the authors.

*Acknowledgements.* Part of this research was carried out at the Jet Propulsion Laboratory (JPL), California Institute of Technology, under a contract with NASA. Susan Kulawik made substantial contributions to the development of the MUSES algorithms and software at JPL.

This research has been supported by NASA via the TRopospheric Ozone and its Precursors from Earth System Sounding (TROPESS) project at JPL and by a Suomi National Polar-Orbiting Partnership (NPP) and the Joint Polar Satellite System (JPSS) Satellites Standard Products for Earth System Data Records grant (no. 80NSSC21K1963) to Karen Cady-Pereira at AER.

This research was also supported, in part, by the U.S. Department of Agriculture, Agricultural Research Service. Mention of trade names or commercial products in this publication is solely for

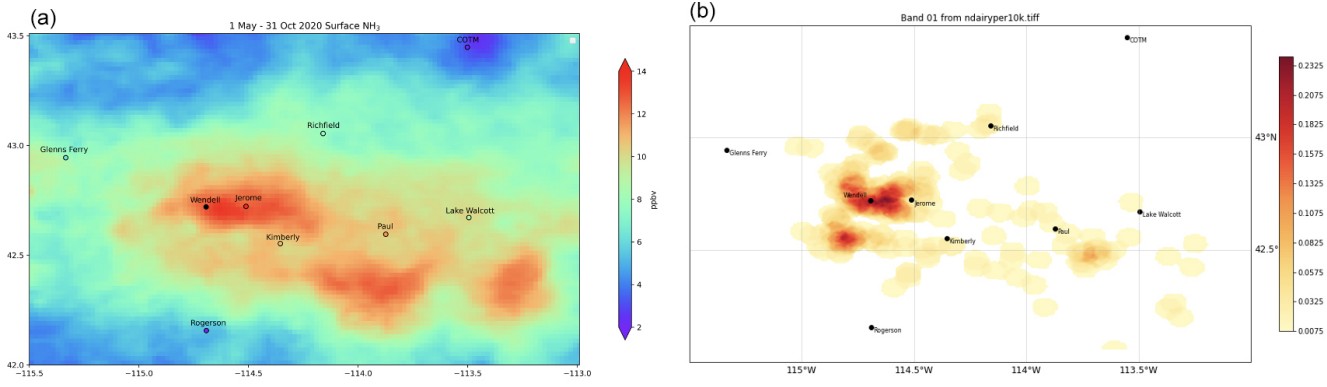

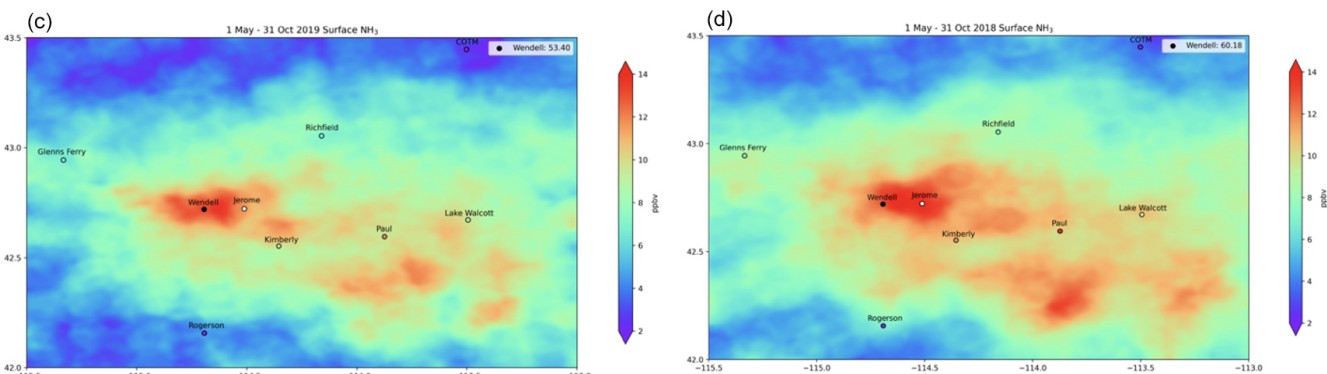

**Figure 12.** CrIS surface NH$_3$ data from the May–October period oversampled onto a 0.002° grid for each of 3 years: 2020 **(a)**, 2019 **(c)**, 2018 **(d)**; number of dairies per square kilometer **(b)**.

the purpose of providing specific information and does not imply recommendation or endorsement by the U.S. Department of Agriculture. USDA is an equal opportunity provider and employer.

NH$_3$ measurements during DISCOVER-AQ were supported by the Austrian Federal Ministry for Transport, Innovation and Technology (bmvit) through the Austrian Space Applications Programme (ASAP) of the Austrian Research Promotion Agency (FFG) (grant nos. 833451, 840086). Tomas Mikoviny is acknowledged for field support; Ionicon Analytik is acknowledged for instrumental support.

Kang Sun acknowledges support from NASA ACMAP (grant no. 80NSSC19K0988).

*Financial support.* This research has been supported by NASA via the TRopospheric Ozone and its Precursors from Earth System Sounding (TROPESS) project at JPL and by a Suomi National Polar-Orbiting Partnership (NPP) and the Joint Polar Satellite System (JPSS) Satellites Standard Products for Earth System Data Records (grant no. 80NSSC21K1963) to Karen Cady-Pereira at AER.

*Review statement.* This paper was edited by Kimberly Strong and reviewed by two anonymous referees.

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

TS3    Please provide the short title, i.e. the shortened form of the main title.

TS4    Please give an explanation of why this needs to be changed. We have to ask the handling editor for approval. Thanks.

TS5    Please give an explanation of why this needs to be changed. We have to ask the handling editor for approval. Thanks.

TS6    Please give an explanation of why this needs to be changed. We have to ask the handling editor for approval. Thanks.

TS7    Please confirm the time. Or should it be changed to "01:30"

TS8    Please advise what should be changed here.

TS9    Please give an explanation of why this needs to be changed. We have to ask the handling editor for approval. Thanks.

TS10    Please also provide the day and month.

TS11    Please also provide the day and month.

TS12    Please provide this information (day/month/year).