# Peer review of "Validation of MUSES NH3 observations from AIRS and CrIS against aircraft measurements from DISCOVER-AQ and a surface network in the Magic Valley"

_Atmospheric Measurement Techniques, 2022_

## Author Comment (AC2)

We thank the reviewer for many insightful comments and good suggestions, especially regarding the point about considering attention the uncertainties of the aircraft data. Our responses to the reviewer's comments are below in blue. Where there were extensive modifications to the text we point to the relevant section rather than including all the modified text in this document. Please note that the reviewer's text appears somewhat oddly formatted (maybe this happened during the conversion to PDF).

Validation of NH3 observations from AIRS and CrIS against aircraft measurements from DISCOVER-AQ and a surface network in the Magic Valley

Overall summary:

I welcome any validation study for ammonia satellite products as, like the authors noted, there are precious few available. Aircraft based measurements provide an almost ideal method for validation which makes this study highly relevant. Even more relevant is the fact that these are, to my knowledge (limited as it is), the first detailed evaluation of CrIS retrieved profiles under inversion conditions. It is also the first study describing the MUSES retrieval and validation. However, some of the approaches taken in the study seem counter intuitive and the manuscript/method will either need some adjustments or at least some further detailing.

Major comments

1. As this seems the first study describing the MUSES retrieval for AIRS and CrIS, that in itself could be given more focus by adding it to the title of the manuscript. Furthermore, illustrate the value by adding a summary to the conclusions on the strengths of the retrieval / improvements over CrIS-FPR.
MUSES was added to the title, a good suggestion. A thorough comparison of the CFPR and MUSES algorithms was not the goal of this paper, but we have added a short section in the Supplement (see next response), where we also highlight some of the differences in the algorithms.

2. As described in the manuscript there is a clear difference between the CrIS-FPR and CrIS-MUSES retrievals as well as the previous AIRS and MUSES retrievals and it should be more clearly described to be as such. The current title and some of the statements ("Preliminary comparisons have shown excellent agreement between the two algorithms") might make it seems that this validation study is applicable to CFPR as well, which is not the case. To not confuse any future reader, please make it more clear what retrieval is used in the study and leave out any comparisons with other products. An alternative option is of course to include the CrIS-FPR retrieval to this validation study to illustrate the differences.
A brief section was added in the supplement describing the preliminary results of a comparison between MUSES and CFPR. This was done at the requests of a several of the

co-authors and a colleague, who felt this was important, given that the CFPR has been widely used.

3. On several occasions, throughout the manuscript, the authors stress the importance of the applying the observation operator for a fair comparison to any second set of observed or modelled concentrations and/to reduce the impact of the apriori choice. While its clear, as stated several times, that the retrievals add information beyond any loss during the apriori choice, the information available is still limited and therefor the apriori will have a large influence. Already in the introduction it's made clear that a comparison of in-situ ground based observations is complicated and highly uncertain due to the strong influence of local atmospheric conditions and vertical distribution! (line 166-168). Several of the comparisons however are made without applying the operator (e.g. Figure 3, Figure 7, and any in-situ observation comparison). Any argument such as "there are many end-users who will want to use the data as is in their own analysis and will want to know the corresponding uncertainties. " or "IASI studies also don't use an operator" are no reason to not apply an averaging kernel and reduce the overall quality of the validation study. If anything it should be stressed once more that comparing satellite observations to in-situ data is not trivial. Assumptions can be made for vertical profiles (e.g. based on apriori shape, modelled profiles, or mixing layer height) after which an AVK can be applied. For the two week averaged concentrations an effective averaging kernel can be approximated. The authors strongly agree that applying the AVK is the optimal method for evaluating satellite retrievals. This was done for Figures 5 and 6, and users carrying out assimilations or inversions will use the AVK and the error covariance matrices in their work. But there will be other users who will simply be using the satellite in lieu of or in addition to in situ data, and will need to be reminded that there are major differences in what these different datasets are actually measuring. The authors feel that presenting the direct comparisons demonstrates these differences. We have revised the text in the MUSES and in the DISCOVER-AQ section to emphasize the importance of applying the instrument operator.

4. Validation vs Evaluation: One could argue that the study is not a validation but evaluation of the profiles as the vertical extent by the flights is limited to either 500 or 700hPa (and only surface for the in-situ obs). The further assumption that all concentrations above those levels are zero doesn't help bring the comparison closer to one another. Please change the title to evaluation and/or make a better assumption for the concentrations above 500/700hPa.

   We understand the reviewer's concerns but believe this is a validation, in that we are comparing the profiles over the mixed layer, the vertical range where almost all the ammonia in the atmosphere is concentrated. While the question of the how much ammonia is actually present above mixed layer is certainly important, especially over fire plumes, as we discuss in the rewritten section on the columns, it can only be resolved with better in situ instruments flying at higher altitudes under different conditions.

5. Which brings us to: The noise in the observed concentrations… make it hard to trust any of the observed concentrations above a certain level (950 (top figure 2) and 800hpa(bottom figure 2)) and concentration value (>5ppbv). Instruments capable of measuring ammonia at high temporal resolution are prone to large bias/artefacts (Bobrutzki et al., 2010/ Twigg et al., 2022) especially at "low" <10ppbv concentrations(funny standard) . The concentrations during the flights show an overall variation of 10 and 4 ppbv depending on the flights/profile/direction etc. This does not match the assumption of 0 ppbv above the measured profiles. While the variations could simply be instruments noise/artefacts, that does reduce the value of all measurements above a certain altitude (950/800hPa). Is it possible to perform any further QA/QC and provide ancillary information, such as the measurement error and mixing layer height to reduce the overall signs that the instrument is simply measuring noise/offset/artefacts.

The reviewer's comment prompted us to reach out to the instrument PI, who informed us that the detection limits for $NH_3$ during DSICOVER-AQ (7.0 ppbv in California and 3.0 in Colorado) were much higher than we had assumed , based on those the PI had published for the VOCs measured during DISCOVER-AQ. Therefore we have rewritten our analysis of the validation stating that it is not possible to make any conclusions about the validity of the satellite data at altitudes where the aircraft data drop below the cited limits. We have added text in in the DISCOVER-AQ introduction and some plots in the Supplement that we hope insight into the variability at each level. We have also pointed out that the aircraft data are binned in layers around each CrIS retrieval level, and that these layers contain hundreds of aircraft measurement, from both ascent and descent.

https://amt.copernicus.org/articles/3/91/2010/

https://amt.copernicus.org/articles/15/6755/2022/

1. Not sure if to place the next comment under minor or major comments:

Line 158-169: A great summary of the pitfalls of previous studies… that we then proceed to walk into in this study. Each of these factors, up to a degree, can be accounted for and improve the validation study:

- Sub-pixel inhomogeneity, a fun point that almost no study does anything with besides mentioning it, some parts from Souri et al., 2022, could help https://amt.copernicus.org/articles/15/41/2022/.

The authors studied the Souri paper and had several exchanges with the author. After some deliberation we determined that the Souri approach, while interesting, would not add useful information the paper. The best approach for estimating the impact of sub-pixel inhomogeneity is to have data that actually measures the variability of NH3 across co-located satellite pixels.  We are doing exactly this with some QC-TILDAS data deployed on an aircraft flown over Colorado last summer during the TRANSAM campaign http://catalog.eol.ucar.edu/trans2am). We are also looking at HYTES data

taken over the Imperial Valley. We now mention these activities in the conclusions, and thank the reviewers for prompting us to revisit this issue.

- Time-scales: the mismatch in representativeness of in-situ networks and satellite will result in a bias. Only a rough statement is made (line 554) but its unclear what the exact value is in this case. With a rough lifetime of 4-12 hours the concentrations that CrIS/AIRS observe will be a combination of emissions over the last few hours and not just the overpass. You could argue that CrIS will be more representative of morning/nighttime emissions and not the peak afternoon values. Please make the potential impact more quantitative by adjusting for the impact or adding a rough uncertainty estimate to the observations.

  This is certainly a valid question. We responded with the following text in the manuscript:

  *While there are strong diurnal cycles in the $NH_3$ emitted from the dairy facilities (Leytem et al., 2011, 2013) the average daily emissions and temperatures, which strongly control the emissions, are close to the early afternoon values. Ideally one would use measurements of the daily cycle in NH3 concentrations, to estimate the ratio between the 13:30 and 24 hour mean concentrations, as was done by Pinder et al. (2011), but such data are not available for the Magic Valley site.*

- Noise of the in-situ or satellite instruments?

  Satellite instrument noise is specified in section 3.3. This translates to actually low error in the NH3 retrievals (3-23%). The much larger components of the error are the smoothing and systematic errors, as well as sampling issues, as discussed in relation to Figure 6 and the new Figure S3. We have also added the following sentence in section 2.1: *For the retrieved profiles used in this study the measurement error ranged from 3.5% to 23%, the systematic errors mainly from 1% to 60%, with a few cases close to 100%, and the smoothing errors from 24% to 130%.*

As stated the horizontal and vertical distribution of ammonia can have a huge impact on the estimated total columns (e.g. factor 2, van Damme 2014). As stated above make an effort to reduce the potential impact or add a factor of uncertainty to the comparison to account for the potential impact.

- https://acp.copernicus.org/articles/14/2905/2014/acp-14-2905-2014.pdf

  Our rewritten section on the column amounts addresses the question of the vertical distribution. As stated in above in the response to the reviewer's suggestion of using one of the approaches in the Souri paper, and in our conclusions, while the issue of sub-pixel inhomogeneity is important, it requires data collected at the sub-pixel level.

Minor comments

1. Abstract lines 35-37 rewrite needed: The way it's currently written, to me, makes it sound like the validation study only represents a very tiny set of conditions and …not important..

2. Be proud of the study, the highly detailed (smaller set) of observations is a strength!
Thank you for the praise. Rewritten as: These are small datasets taken over high source regions under very different conditions: winter in California and summer in Colorado

3. Line 35-40 add quantities to the bias/errors.
Done

4. In several sections of the manuscript there are statements of outcomes of other validation studies but it is not clear which retrieval was validated. There have been several IASI products over recent years with large difference between them (typically updates). Please add the version numbers.
Done.

5. Line 67: "within the European union" how its currently written makes it sound like the EU regulate US pollutants.
Rewritten as: Ammonia emissions are regulated by the European Union (EU) and it is a criteria pollutant in Canada, but not yet in the US. However he EPA has published established regulations

Line 85: add reference to "measure accurately", for example Bobrutzki / Twigg, https://amt.copernicus.org/articles/3/91/2010/
We have added a reference to the von Bobrutzki paper and modified the relevant sentence: However, in situ measurements remain a challenge. $NH_3$ is easy to detect, but it is hard to measure accurately, especially for concentrations below 10 ppbv (von Bobrutzki, 2010)

https://amt.copernicus.org/articles/15/6755/2022/

1. Line 91: there are several instruments measuring via an open-path that are used in measurement networks (e.g. https://amt.copernicus.org/articles/10/4099/2017/amt-10-4099-2017.pdf, https://amt.copernicus.org/articles/16/529/2023/amt-16-529-2023.html)
Added two references to open path instruments.

2. Line 100-105 add the spatial coverage/footprints of the individual sensors.
This information is provided for AIRS and CrIS in the section 2.2. In this section the authors feel it is not useful.

3. Line 120: reference? (e.g. https://acp.copernicus.org/articles/22/6595/2022/acp-22-6595-2022.html, https://acp.copernicus.org/articles/22/951/2022/acp-22-951-2022.html)
   Provided a direct reference to the AMoN site.

4. Line 125 onward: add version numbers and retrieval names.
   Done.

5. Line 125 onward: instead of good/high/etc add quantities
6. Line 141 what was the result for IASI(-NN and -LUT)?
   Done.

7. Line 142:144: Incorrect statement. Most of the FTIR sites are located **away** from high source regions, which limits the applicability for high concentration regions. Several of the NDACC sites however (e.g. Hefei, Mexico city, Bremen, Boulder) are within or near regions with high concentrations which makes the complete network quite applicable, and to great interest of the air quality community.
8. Line 143: What is of greatest interest to the air quality community?

   The reviewer is correct, this statement is not right and was deleted. Note that only seven of the NDACC sites were used in the Dammer analysis, and that the CrIS Mexico City data were suspect. The following sentence was added to provide more context.
   Correlations at the individual sites ranged from 0.28 (Mexico City) to 0.86 (Bremen).

9. Line 158: Move a part of the sub-pixel bit (169-180) above this section for better readability.
   Done

10. Line 180-182: again makes it sound like the study is not that relevant, while it is!

    Rewrote as : Aircraft campaigns are valuable in that they profile the vertical distribution of $NH_3$, allowing us to evaluate the performance of retrieval algorithms and to provide models with more realistic profiles; however they are by nature limited in their temporal coverage

11. Chapter 2 & 3.3 integrate together into a MUSES chapter
    Done.

12. Line 199: Can you add an example of the apriori profiles and typical surface values.
    A priori profiles have been added as Figure S1. Figure 2 provides good examples of the range of typical values. We also added the following in the introduction: (surface values can range from less than 0.1 ppbv to 200 ppbv or more).

13. Line 214-218: either add a section comparing the two retrievals, provide a source for these results or remove this section.

14. Line 216: specifically "excellent agreement", while I understand the comparison is beyond this paper at least specify where anyone can find the comparison.
    The authors have revised this statement (see also the authors' response to the reviewer's second major comment).

15. Line 256:261: please add some quantities (ppbv/%) to what can be expected for each of the error sources. Show that III in particular is essential as it seems the part that's new within the MUSES retrieval.
    As stated above, the following was added to section 2.1:
    For the retrieved profiles used in this study the measurement error ranged from 3.5% to 23%, the systematic errors mainly from 1% to 60%, with a few cases close to 100%, and the smoothing errors from 24% to 130%. Example retrieved profiles and corresponding errors are shown in Figure S3.
    We also added the following at the end of the paragraph: Note that the estimated error cannot account for sampling errors, i.e., differences between the air masses sampled by the satellite and by the in situ instruments.

16. Line 284-287: Either remove, or add a few lines/statement to the discussion/conclusions that this study indicates the potential hazards of, and large levels of uncertainty in, simply using the CrIS retrieved surface concentrations.
    Please see response to third major comment.

17. Line 290: After the whole description, simply truncating the averaging kernel seem counter-intuitive. Of the limited information contained in each observed profile quite a bit will be above the 500/700hPa level. Smoothing etc will have an effect on the profile/column comparison. Please show that this effect is minimal or (better) redo the comparison without truncating and assume a value for the levels above 500/700hPa.
    We followed the reviewer's suggestion and extended the aircraft profiles by blending in the MUSES NH3 prior above the top of the aircraft data and redid the profile comparisons, statistics table and error plots, then adjusted the text accordingly. The differences between applying the full AK to the extended profiles and applying the truncated AK to the aircraft only data were quite small.

18. Line ~295: add the dates/period to California 2013 and Colorado 2014.
    Dates were added, and the text was reorganized.

19. Line 310-311, why not bin the observations the CrIS and AIRS retrieval intervals?

    The values in Figure 1 were binned for clarity in the plot, which is just an illustration. For the comparisons with AIRS and CrIS, the aircraft observations were binned on the retrieval intervals, as stated in section 4.

20. Line 311: add the detection limit, and concentration interval that the 35% is representative of.
21. Line 314: "higher detection limit" how much higher?
22. Line 315: same, what amounts are we talking about?

After consulting with the PTR-MS instrument PI this section was rewritten as:

*Note that the PTR-MS instrument samples the atmosphere at 1Hz but the data in Figure 1 were binned over 100 m to improve visibility. The estimated instrument uncertainty is 35% (Müller et al., 2014). However, the PTR-MS $NH_3$ data were a side product of the PTR-MS measurements during DISCOVER-AQ, which were designed to obtain data on volatile organic compounds (VOCs). $NH_3$ is sticky and accumulates in the instrument inlet, slowing the instrument response This effect leads to biases if the $NH_3$ amounts are changing rapidly (Sun et al., 2015);when the aircraft is leaving the boundary layer on upward spirals the instrument does not respond quickly enough to the sharp decrease in $NH_3$ and overestimates the $NH_3$ concentration; similarly, when entering the boundary layer on downward spirals, the response to the increase in $NH_3$ is slow, and $NH_3$ is underestimated (see Figure 9 in Guo2021). Furthermore, the detection limits for $NH_3$ were much higher than for the VOCs that were the primary target of the PTR-MS measurements: 7.0 ppbv in California and 3.0 ppbv in Colorado (Armin Wisthaler, personal communication). These limits imply that any aircraft observations below these values are effectively noise.*

23. Line 323: technology and protocols: add reference.
24. Line 325 type of sampler? Quite the quality differences.
25. Line 346: add LSTs.
    Added.

26. Line 351: if CrIS-JPSS-1 is not used leave it out of the manuscript.
    We respectfully disagree on this point and have listed both JPSS-1 and JPSS-2, since $NH_3$ from CrIS on these platforms will greatly extend the $NH_3$ record.

27. Line 361: add some information on why 60 minutes and 15km are used (especially for low(<5kmph) and high (>30kmph) wind speeds differences in observed air mass are possible.
28. Line 362: summarizes the approach of Guo2021.
29. Line 363-364: stricter for a reason, other studies (e.g. Tournadre, 2020 for NH3) showed some information on the limits, reference.
https://amt.copernicus.org/articles/13/3923/2020/amt-13-3923-2020.pdf

This section was rewritten as shown below. The authors hope this addresses the reviewer's concerns.

The DISCOVER-AQ campaigns in California and Colorado provide the most comprehensive set of in situ $NH_3$ profile data (as opposed to retrievals from FTIR instruments) available. Both locations have many strong sources and each campaign carried out multiple flight days over a two month period. These datasets demonstrate the strengths and limitations of satellite data in areas of great interest to the air quality community; additionally, they allow for the evaluation of the accuracy of the retrieval estimated error, as calculated from Equation 3. During each flight the aircraft flew multiple up and down spirals. The satellite profiles were co-located with aircraft profiles taken within one hour of the satellite overpass time and 15 km of the pixel center, the same criteria used by Guo2021.This co-location criterion is much stricter than is usual for satellite validation (see Hegarty et al., 2022 for an example with CO retrievals from AIRS, who used nine hours and 50 km) but is necessary given the short lifetime of NH3 (on the order of hours to days) due its high reactivity and fast deposition. Tournadre et al. (2020) used an even stricter time requirement of 30 minutes for comparing FTIR and IASI NH3 retrievals over Paris, but we found that such a limited time window drastically reduced the available data. Given the chosen criteria, each CrIS or AIRS profile was compared with data from at most two spirals

1. Line 365-367: add values for future reproducibility
   Done.
2. Line 369: Why median and not mean? Did the data have strong outlier values / not well distributed?
   Yes, roughly 25% of the profiles had very long tails. Please see Figures S4 and S5.

3. Line 372-377: move up to line 347 as its appropriate to already mention the differences there.
   Done

4. Line 382: "amounts as high as 100ppbv …" add during a (weak) inversion?
   NH3 amounts downwind from CAFOS routinely reach values above 100 ppbv, even when there are no inversions (see Nowak et al., 2012).

5. Line 385:390: excellent case-study no negativity/toning down needed, adjust text.
   We thank the reviewer for this suggestion and have rewritten this section as: There were thermal inversions over the entire period (Figure 2, upper right), which lead to increased uncertainties in the retrieval, as they effectively create an emission layer above the surface, i.e., a layer that is warmer than the surface and therefore emits more than it absorbs. Inversions also limit the vertical extent of the boundary layer, with consequently lower $NH_3$ concentrations at altitudes where the retrieval has greater

sensitivity. Nevertheless, evaluating the AIRS and CrIS $NH_3$ profiles against the aircraft data is a useful exercise, as the combination of inversions and strong sources is not a rare occurrence, and this analysis demonstrates both the capabilities and limitations of retrievals under these conditions.

6. Line 391: "However, when averages over long periods and/or broad regions are desired, it would be reasonable to exclude cases with inversions", I have to disagree and argue the opposite, and ground-based measurement will also measure during these inversions, for an accurate comparison you'll need to include such events into the satellite mean, else its not representative of the situation on the ground.
   This is an excellent point and we have removed this sentence.

7. Line 395-407, to be honest this whole section could be removed. From previous studies and your summary in the introduction its already clear that there are large bias (or representation errors) to be expected from not applying the averaging kernel or spatial-heterogeneity, its not needed to show it here.
   Please see our response to the reviewer's third major comment.

8. Line 426: Please add the percentage that the 1.0ppbv represents compared to the total observed concentrations or add those to the text.
   Added the following to the text: roughly 7% to 10 % at the surface, increasing to 30% at 750 hPa

9. Line 427-429: Importance was already stated in the introduction. If you want to leave this section in, add some colouring of the profiles (fig3,fig5) based on the apriori profile concentrations. A percentage based plot would also help put the values into perspective.
   We are not sure exactly the importance of what the reviewer is referring to, but we have taken the suggestion of coloring the profiles and the column amount by the choice of the a priori profile.

10. Line 436: …has been argued…add a refence.
    We apologize; this statement was made to us at several conferences, but never in a paper. We have removed the sentence.

11. Line 451: similar like stated above, add some values on what order of uncertainty/error/bias we can expect for the individual errors. A plot like Fig. A2/A3 in Dammers et al., 2017, comes to mind. https://amt.copernicus.org/articles/10/2645/2017/
    A plot following Dammers Figure A2/A3 would be interesting but would require extensive analysis and is beyond the scope of this paper.

12. Line 453-455: "The measured uncertainties range from 5%-50% …. point to the need for averaging…" why is that the case? Most in situ instruments/measurements observe with comparable levels of uncertainty, similarly the uncertainties in the emissions can be up to several orders (Factor 2.5, higher values also mentioned within Van Damme 2018, stated in introduction / Dammers 2019 also in introduction).
    We agree with the reviewer that this statement is incorrect and have removed it.

13. Line 459:461: Again not an argument to make the same (incorrect) comparison here. Either replace entirely with, or add a comparison including the application of the averaging kernel and show the impact on the comparison.

We have experimented extensively with different approaches for carrying out the total column comparisons: using all the aircraft data, extending the aircraft data to TOA, as was done for the reworked Figure 5, applying and not applying the AK, and every case yielded significantly worse results for the AIRS Colorado comparisons and lower correlations for all cases. The figure below shows what happens when we integrated the reworked profiles from Figure 5 (aircraft profiles extended to TOA with the AK applied).

[Figure]

We leave it to the reviewers to decide if this plot is useful. Instead we truncated the AIRS/CrIS profiles to the MLH and compared those to the aircraft columns integrated to the MLH (new Figure 8). We also extensively reworked the text on the column comparisons for both California and Colorado, discussing the role the choice of the a priori profile. We feel comparing the total and partial columns, and considering the effect of the a priori choice and the aircraft uncertainties, along with the fact mentioned by the reviewer that July and August are fire season in Colorado, has provided some useful insights.

14. Line 461: incorrect statement, Dammers et al., 2017 (and 2016) did apply the averaging kernel to IASI profiles. The IASI retrieval uses a profile assumption and profiles can be derived from the columns. https://acp.copernicus.org/articles/16/10351/2016/

The IASI retrieval product consists of columns. Profiles can be derived from columns in a post-processing step, as Dammers et al., 2016 and van Damme et al., 2017 have done, but in both papers just two fixed profiles were used to convert from column to profile. The authors believe that this process adds uncertainty to a comparison with already large inherent uncertainties. But we expanded our statement by adding: *though Dammers et al., (2017) estimated IASI columns by using two fixed vertical profiles to convert column amounts to profiles.*

15. Line 469: "are assumed to be zero" As stated above this is quite an assumption and the impact should be quantified. Alternative choices such as using the values from the apriori profile or scaling the apriori profile with the observed values at the top of the spiral are also viable. The lifetime is of the order of hours – days which means there should be a non-insignificant amount of ammonia above the mixing layer and in the upper

troposphere. The July-August measurements in Colorado coincide with the fire season, long-distance high-altitude plumes could occur during this period and interfere in the comparison (e.g. Lutsch et al., 2016; https://agupubs.onlinelibrary.wiley.com/doi/full/10.1002/2016GL070114).

16. Line 486:487 + beyond: At the altitudes where the measured concentrations seem valid enough there is no indication of a missing error source. If the values of the aircraft are not representative of concentrations observed >500/700 hPa how can we still conclude anything about potentially missing error sources?

    See our response to Major Comment 5. Based on our revised understanding of the validity of the data the reviewer is entirely correct that there does not appear to be any unknown error source in the Colorado dataset.

17. Line 515: …water vapour retrieval errors… Add a bit of discussion on this outcome as the sub-pixel retrievals of water vapour etc were one of the addition of MUSES over the old retrievals (unless I am mistaken, always possible). If this only increases the bias/uncertainty is it smart to keep doing the retrievals as such?
    The authors do not consider the water vapor retrieval an addition to the process. Any optimal estimation retrieval algorithm for NH3 (or any other minor trace gas) requires information on the atmospheric state. The CFPR and MUSES algorithms simply obtain the atmospheric state in different ways. The CFPR algorithm uses the NUCAPS water vapor and temperature derived from cloud cleared CrIS radiances (on the coarser nine pixel resolution). The MUSES algorithm starts from the single pixel radiances, and over multiple steps, retrieves surface properties, cloud optical depth, temperature and water vapor, as well a number of different trace gases (CO,CH4,O3) besides NH3. This process ensures that the atmospheric state is derived using the same forward model and radiance data that are used in the NH3 retrieval, reducing possible sources of error. The last statement has been added to the MUSES description.

18. Line 545: biased high should be biased low. Puchalski et al also gave a range of +18 to - 32%.
    We thank the reviewer for pointing out this error and have corrected it.

19. Line 554: see earlier statement on using emissions for concentration representativity.
20. Line 559: "possibly" missing a closing bracket
21. Figure 9: Excellent example of the temporal variability picked up by the satellite instrument However, accounting for the apriori effects could bring these comparison a lot closer.

    Yes, maybe. However, this would require introducing a great deal of extra data (profiles above the surface, averaging kernels), which we suspect would just muddy the comparisons, as we found when we extended the DISCOVER-AQ profiles and applied the AK.

22. Figure 11: colorbar values are missing
    Corrected.
23. Line 616: add within the "United States" or something similar, recent EU emission databases are typically at distributed based on livestock numbers at each farm, with some inventories (e.g. the Netherlands) even at facility level that get aggregated to $1x1km^2$
    Done.

24. Line 621: oddly specific, leave out, or add reference? (Herrera, 2022, https://acp.copernicus.org/articles/22/14119/2022/acp-22-14119-2022.html)
    We thank the reviewer for reminding us to cite the paper we were thinking of.
25. Line 624-626: Be proud! You're reducing the relevance of this paper, while it definitely is relevant!
    We believe we have addressed this issue throughout the paper and thank the reviewer for the excellent suggestion.

26. Conclusions: update statements following the above comments.
    Conclusions have been updated.

27. Line 624: show the shape and values in the retrieval section.
    We are not sure here what the reviewer referring to here.

---

## Author Comment (AC3)

The authors thank the reviewer for some insightful comments and suggesting areas that require further discussion. We have attempted to address all the reviewer's comments below (in blue). When the text required significant revisions we pointed to the section of the updated manuscript where the revisions were made. We have included some of the reviewer's suggestions in section 6 (Conclusions and future work).

The present study describes the validation of ammonia ($NH_3$) data retrieved from the AIRS and CrIS satellite instruments using the MUSES algorithm. The study focuses on comparing $NH_3$ profiles and columns derived from aircraft measurements obtained during the DISCOVER-AQ campaign in California and Colorado, specifically in source regions of ammonia. Additionally, it includes a comparison with three years of surface $NH_3$ measurements from a monitoring network in Idaho.

The manuscript is well written, properly structured, and aligns well with the scopes of AMT. It contributes to the extensive validation efforts of various $NH_3$ products derived from satellite measurements, which is crucial considering the growing utilization of $NH_3$ satellite data. There is an evident need for such validation. Overall, the comparisons between satellite and in-situ data are well executed and yield interesting results.

However, I have some general concerns regarding the significant uncertainties inherent in the comparisons involving satellite-derived mixing ratios and in-situ measurements at specific altitudes/pressure levels, particularly at the surface. This concern arises due to the absence of vertical information that can be obtained from trace gas retrievals like $NH_3$. Additionally, I believe the manuscript lacks thorough discussion and investigation into the reasons behind the remaining biases/uncertainties. Addressing these aspects would provide valuable material for the paper's conclusion.

Therefore, I recommend publication once the following major comments listed below are addressed.

**Major comments**

A major conclusion of the study is that a portion of the biases between satellite and in-situ measurements can be attributed to smoothing errors and unaccounted error sources, which can be substantial. This is evident in Figure 5, where the standard deviation of the biases remains large even after applying the AVKs. However, I believe the study could go deeper into understanding the sources of these biases and uncertainties, and additional tests within this framework would be beneficial. Specifically:

- The manuscript should discuss the uncertainties and errors associated with temperature and $H_2O$ profiles, as they often have significant impacts on trace gas retrievals from nadir-viewing satellite observations, particularly for $NH_3$ that mainly resides in the boundary layer. It would be helpful to explore whether the remaining biases between satellite and in-situ measurements are dependent on errors in these two variables.

Additionally, what is the influence of uncertainties in temperature and H2O profiles on the overall NH3 uncertainty budget?

This is certainly an important topic, and prompted by the reviewer's question on why systematic errors were not included we reviewed the text and realized this had been stated incorrectly and has now been now fixed. The MUSES NH3 retrieval step includes the estimated errors in NH3 due to water vapor and temperature errors in the systematic error. We now mention the range of systematic errors in the text. We also state the estimated errors in Figure 6 are sum of the measurement error and the systematic error, and that since the measurement error is usually small, these estimated errors are nearly equivalent to the errors due to water vapor and temperature errors. Given the good agreement between the estimated errors and the actual uncertainties in Colorado below the MLH, we believe these uncertainties have captured the error sources. In California it is obvious from Figure 6 that the retrieval errors are underestimated, either because the systematic errors are incorrect or there are missing error sources. We have also added the following to section 6.

*This study has not attempted to untangle the impact of the errors in the retrieved water vapor from those of temperature on the $NH_3$ errors.. Such an analysis is an important and ongoing task, as global maps of CrIS NH3 have revealed artificial hotspots of NH3 over tropical oceans, where humidity is high. There is a weak water vapor line in the spectral region used in the NH3 retrievals, which is possibly leading to these artifacts.*

- Information about the DOFS associated with the AIRS and CrIS NH3 retrievals would be interesting to discuss. Did you exclude observations based on their DOFS before conducting the validation? Do the biases between satellite and in-situ measurements decrease when filtering out observations with low DOFS? Exploring this aspect would provide valuable insights.

  The small number of aircraft profiles led us to exclude only truly poor retrievals. We have modified the text where we discuss the QC as follows:

*Retrievals were checked for quality by ensuring that for all retrievals the root mean square error (RMSE) of the residuals was less than 5.0. The MUSES cloud optical depth (COD) values were also evaluated but since the maximum COD for the retrieved profiles was 0.25 no retrievals were rejected due to large COD. Four CrIS profiles over Colorado were rejected due to very high estimated uncertainties. The DOFs ranged were between 0.8 and 1.1, except for two CrIS profiles over California, four AIRS profiles in California and six AIRS profiles in Colorado, for which the DOFS were smaller (0.2 to 0.7). Given the small number of profiles in each dataset, we did not exclude any profiles based on the DOFs.*

- The choice of the a priori profile is crucial. Are the selected a priori profiles representative enough for the conditions in California and Colorado during the in-situ measurements? How would the biases between satellite and in-situ measurements change if you adopted an a priori profile that peaks closer to the surface? Would such a choice help reduce the biases? This is an aspect worth investigating and discussing.

  The authors are well aware of the impact of the a priori on the retrieved profiles. In the column comparisons in section 4.1 and 4.2 we now discuss how the choice of a background or moderate a priori (see new Figure S1 in the supplement) leads to a very different vertical distribution than the enhanced prior, which does peak at the surface. Comparing the a priori profile shapes with the visualization of the aircraft profiles shown in Figure 2 suggests that over California the enhanced a priori profile reflects the shape of the measured profiles, albeit with a shorter tail and more gradual rise. Over Colorado an enhanced prior with simply a steeper rise would have been ideal. These modified a priori profiles might have led to better agreement with the aircraft data, though please note the new discussion about the detection limits of the PTR-MS over California (7.0 ppbv) and Colorado (3.0 ppbv). However, using modified a priori profiles would have defeated the purpose of this paper, which is to evaluate the MUSES retrievals as they are now and to provide users who have utilized these retrievals an estimate of the current uncertainties. In the next phase of the algorithm development we will experiment with modified retrieval shapes and the addition of a super-enhanced prior with a long tail. We have taken the reviewer's suggestion and added this issue to the discussion of future work in section 6.

- I have concerns when comparing trace gas mixing ratios retrieved from ground-based or spaceborne observations at a specific altitude/pressure level directly with in-situ measurements taken at the same level. It's important to recognize that, with the optimal estimation, the value retrieved at this level alone lacks meaning as it heavily relies on information obtained from other levels. This is particularly relevant for trace gases like NH3, where only a single piece of information (total column) can be obtained. It becomes even more complex when comparing surface in-situ measurements with near-surface mixing ratios derived from retrieved profiles, considering the decreased sensitivity of satellite IR sounders in the lowermost tropospheric layers. Although the sum of each row of the AVKs shows some sensitivity to the near surface, the influence of the a priori profile remains substantial in these layers. Furthermore, the shapes of the AVKs indicate a tendency for the retrievals to overcompensate in the free troposphere (Figure 4), suggesting that part of the information used to estimate near-surface values originates from higher levels. Additionally, I assume that the constraint matrix restricts the variability in these layers, incorporating inter-layer correlations through the extra-diagonal elements, in order to prevent abnormal oscillations in the retrieved profile and maintain it within a reasonable range relative to the a priori. Considering these factors, it is crucial to thoroughly investigate and discuss the extent to which these retrieval characteristics impact the comparisons between satellite and in-situ data before drawing conclusions.

*The authors agree with every point the reviewer has made here.  Yes, the constraint matrix was designed to restrict variability in the free troposphere. We have added the text below in the section describing the SRAK, and added further comments in the column section on the impact of the a priori choice.*

*The sum of the rows of the averaging kernels (SRAK) (Figure 4, top two panels), which provides an estimate of the retrieved information at each level originating from the measurement rather than from the a priori, shows for both AIRS and CrIS that while the information from the radiance data peaks just below 700 hPa, it  also significantly contributes to the retrieved surface values. This is driven by the structure of the covariance matrix ($S_a$). As noted in the introduction of the DISCOVER-AQ section, the DOFS for AIRS and CRIS NH3 ranged mostly between 0.8 and 1.0, signifying the retrieval provides only one piece of information, basically a column amount. By building off-diagonal correlations in a priori  covariance matrix between the surface level and a few levels above, this information is vertically distributed in such a way that it restricts unphysical oscillations in the retrieved profile and deviations a priori profile shape. Each of the three a priori profiles is associated with a different covariance matrix. The enhanced a priori retrieval tends to load the profiles at the surface, while the moderate and background profiles push NH3 to the free troposphere*

In Guo et al. (2021), it was demonstrated that notable differences exist between the ascent and descent aircraft profiles of NH3 measurements obtained during the DISCOVER-AQ campaign. These profiles are utilized for validating the AIRS and CrIS NH3 observations in the current study. The observed disparities arise due to the slow response time of the PTR-MS instruments, leading to a sampling lag when the aircraft moves from the boundary layer to the free troposphere, and vice versa. Since the majority of NH3 is concentrated in the boundary layer, this response lag results in an overestimation of NH3 in the free troposphere during upward spirals and an underestimation of NH3 in the boundary layer during downward spirals. These biases between ascent and descent profiles have been identified as significant and, when combined with the 35% uncertainties associated with NH3 measurements, can considerably impact the comparisons with satellite measurements. However, this point is not discussed or accounted for in the present study.

*The authors agree that Guo et al. (2021) demonstrate this effect very clearly. We have added the following text that we hope addresses the reviewer's concerns:*

*The CrIS retrieval layers are fairly coarse and therefore the median value of the PTR-MS is derived from a set of hundreds of measurements spanning the layer, and from both up and down flight paths, thus possibly reducing to some degree the biases from entering and leaving the boundary layer discussed above.*

*However, since the initial draft of this paper was submitted to AMT we learned we had assumed the wrong detection limits (based on Müller et al. 2014) for NH3 during these campaigns, and the true limits were much higher: 7.0 ppbv in California, and 3.0 ppbv in Colorado.  Examining the PTR-MS data we found that basically almost all the observations above the MLH were below these limits and effectively noise.*

In the comparison of NH3 columns between satellite and in-situ measurements (Figure 7), it is evident that both AIRS and CrIS tend to overestimate NH3 columns in California and Colorado, as indicated by the slopes ranging from 1.6 to 2.3. However, the application of AVKs has not been performed on the aircraft profiles in this comparison. Since satellite sounders exhibit reduced sensitivity and obtain less information in the near-surface layers, where NH3 is predominantly abundant, it is expected that the AVKs would diminish the influence of these layers when computing NH3 total columns from the smoothed aircraft profiles. Consequently, the aircraft NH3 columns may be lower, potentially accentuating the overestimation of NH3 by AIRS and CrIS. This aspect should be discussed.

The authors have invested considerable effort trying to understand the total column results. We did try calculating total aircraft columns by integrating the extended profiles

with the AK applied (profiles shown in the new Figure 5), but found the results difficult to understand (see plot to the left). In fact we tried a number of different approaches with and without applying the AK and in the end opted for a simple experiment. Since the aircraft data above the MLH is unreliable, we continued to use the aircraft data integrated to the MLH, and did the same for the AIRS/CrIS data (new Figure 8). We feel that comparing the total and partial columns, and considering the effect of the a priori choice and the aircraft uncertainties, along with the fact mentioned by the reviewer that July and August are fire season in Colorado, has provided some useful insights. Please see the text in the column section for our conclusions.

On the other hand, in section 5.2, it is revealed that CrIS NH3 exhibits an overall low bias when compared to surface in-situ measurements (slope of 0.65). This finding contradicts the previously noted large overestimation observed for CrIS NH3 columns (as mentioned in my previous comment). This discrepancy raises additional concerns regarding the reliability of quantitative comparisons between surface in-situ and satellite data. It suggests that such comparisons at the surface level are particularly uncertain and prone to biases.

Please keep in mind that Figure 7 (and the new Figure 8) compares columns, while Figure 11 compares surface values. In the section on columns the authors now discuss why the AIRS/CrIS total columns are biased high with respect to aircraft columns, which only include reliable observations up to the MLH. Columns can be high while the surface values are low because the retrieved vertical distribution does not match the true vertical distribution, as the reviewer has also noted. So yes, the surface level retrievals are prone to biases, but a large fraction of in situ data is surface data, so it is worthwhile doing the comparisons. The authors also feel that the time series in Figure 10 show little bias in the warm season; in cold weather not only are emissions

lower, and therefore NH3 concentrations are closer to the AIRS/CrIS detection limit (0.5 to 1.0 ppbv), but lower temperatures lead to lower thermal contrast and weaker signals..

The study identifies sampling differences between satellite data and in-situ measurements as another significant source of uncertainties and biases, particularly following the application of satellite AVKs. To gain further insights into this aspect, it would be beneficial to conduct tests on the co-location criteria in terms of both spatial and temporal alignment. For instance, reducing the co-location time to 30 minutes could improve the representativeness of the satellite measurements with respect to the in-situ data. Conversely, extending the co-location time would provide a larger statistical dataset for the comparisons, enabling a more robust analysis of uncertainties and biases.

The authors considered this request, but given that GUO2021 had already shown that a one hour and 15 km window was the most appropriate one for Colorado, we decided to adopt that window for Colorado, and for consistency sake, also for California.

**Minor comments / typos**

- Lines 226-228 and 399-401: I find it unclear whether the retrievals are conducted over cloudy scenes as well. If retrievals are performed over cloudy areas, it could pose a concern since the presence of clouds can impact the baseline temperature of the spectra and the thermal contrast, consequently affecting the retrieved column. These effects should be taken into consideration, as they have the potential to introduce biases and uncertainties in the NH3 retrieval process.

  As stated in the text, MUSES retrieves cloud optical depth in a previous step, and this optical depth is included in the radiative transfer calculations for the subsequent steps, including NH3. The maximum cloud OD for the set of DISCOVER-AQ profiles was 0.25, and in general the cloud OD was less than 0.1. In the Magic Valley there were more cloudy days; in the correlation plot we only included pixels with COD less than 1.0, while in the time series we allowed retrievals with COD up to 2.0, in order to highlight the seasonality.

- Lines 263-264: Why are the systematic errors not evaluated for CrIS?
  See response to first major comment.

- Lines 288-290: I believe truncating AVK matrices in such a manner may not be appropriate. It implies that the influence of the upper layers on the remaining layers is entirely disregarded, consequently affecting the smoothing of the aircraft profiles. A possible workaround could involve complementing the aircraft profiles at the top with additional information, such as model profiles scaled to background NH3 values, and applying the complete AVK matrices. Although this approach would introduce an impact from the profile at the top, it seems more reasonable to me than truncating the AVK matrices entirely.

We no longer truncate the AVK. We have extended the aircraft profiles to the TOA by blending the in corresponding AIRS/CrIS a prior profile above the aircraft. The new versions of Figure 5 and Table 1 reflect this change, but note that the change is small.

- Line 292: Müller
  Corrected.

- Lines 350-351: Why are the observations from CrIS/JPSS-1 not used here? They could have filled the gap of CrIS/SNPP in 2019. And it would have been interesting to check the consistency between these two CrIS instruments for NH3.
  The authors considered the reviewer's suggestion, but after some debate concluded that adding the CRIS/JPSS-1 data would have required some substantial processing and would not have significantly changed the results of the Magic Valley analysis. Comparing SNPP and JPSS NH3 is a task for another paper, which would do so globally.

- Lines 365-361: Could you be more specific on the filters you applied to the retrieved data?
- *Retrievals were checked for quality by ensuring that for all retrievals the root mean square error (RMSE) of the residuals was less than 5.0. The MUSES cloud optical depth (COD) values were also evaluated but since the maximum COD for the retrieved profiles was 0.25 no retrievals were rejected due to large COD. Four CrIS profiles over Colorado were rejected due to very high estimated uncertainties. The DOFs ranged were between 0.8 and 1.1, except for two CrIS profiles over California, four AIRS profiles in California and six AIRS profiles in Colorado, for which the DOFS were smaller (0.2 to 0.7). Given the small number of profiles in each dataset, we did not exclude any profiles based on the DOFs.*

- Figure 2: It might be useful to superimpose the mean and standard deviation of the profiles on these plots, as it was done for Figure 3.
  This was a very useful suggestion: the mean and standard deviation have been added to Figure 2

- Line 437: "*It has been argued*". As is, it sounds a bit weird. Do you refer to a specific study?
  We apologize; this statement was made to us at several conferences, but never in a paper. We have removed the sentence.

- Line 477: What is the typical detection limit of NH3 for these satellite instruments?
  These limits have been added to the text:
  *The MUSES total columns are compared against the integrated aircraft columns, using orthogonal linear regression (Figure 7a and 7b); the intercept has been allowed to vary, as both AIRS and CrIS have detection limits, (~1.0 ppbv, for thermal contrast above 5K), as does IASI (3.0e15 molecules/cm2, for thermal contrast above 5K).*

- Lines 482-485: I don't understand what is meant here.
  This section has been extensively rewritten. Please refer back to the new text.

- Line 495: "some poor quality CrIS retrievals". It is surprizing, since CrIS has much lower instrumental noise compared with AIRS.
  Four CrIS retrievals over Colorado had very large estimated errors, due to large systematic errors (from the water vapor and temperature retrieval steps). We have modified the text to make this clearer.

- Line 535 and line 545: There is likely a full stop punctuation missing at the end of each of these two lines.
  Corrected, thank you.

- Line 559: Could the winter values be underestimated because of the general weaker thermal contrast (measurements closer to the detection limit)?
  Yes, thermal contrast can certainly play a big role. We have rewritten this section as: *At every site, CrIS clearly captures the seasonal cycle, though winter values are usually underestimated: this can be attributed to weak radiative signals due to low temperatures and low thermal contrast. The CrIS level of detectability is normally cited as ~1.0 ppbv (Shephard and Cady-Pereira, 2015), but at low thermal contrast this level increases significantly.*

- Line 559: *"(possibly,"*? There might be a typo here.
  We apologize but could not find the typo referred to here.

- Lines 587-588: "*at the high end of the values reported in the literature*" Please provide references.
  We added a reference to van Damme et al., 2015b, and pointed back to the introduction.

- Figure 10: The shapes/limits of the subplots are not consistent between NH3 data vs. number of dairies. Please consider using the same projection. Also, the values of the NH3 colour bars are hidden.
  Figure 10 is now Figure 11. Color bars are now visible. We have tried to make the projections as close as possible.

- Line 631: delete the full stop punctuation after "measurements"
  The text in this section has been changed significantly and this sentence is no longer present.

---

## Author Response (AR2)

The authors wish to thank the reviewer for the effort of carefully reading the our manuscript for a second time. We fixed all the errors the reviewer listed (see below):

- Lines 87-88: "T here"
- Line 353: "in Colorado in July and August 2014 in Colorado in 2014"
- Line 395: "due its high reactivity"
- Lines 421-422: "as as the most…"
- Line 584: "to currently to determine"
- Line 599: "the bulk of the bulk of the NH3"

We also thank the associate editor for recommending larger fonts on some of the figures. Accordingly we revised Figures 2, 4, 5 and 6.